# CAN TRANSFORMERS PERFORM PCA ?

## ABSTRACT

Transformers demonstrate significant advantage as the building block of Large Language Models. Recent efforts are devoted to understanding the learning capacities of transformers at a fundamental level. This work attempts to understand the intrinsic capacity of transformers in performing dimension reduction from complex data. Theoretically, our results rigorously show that transformers can perform Principle Component Analysis (PCA) similar to the Power Method, given a supervised pre-training phase. Moreover, we show the generalization error of transformers decays by $n^{-1/5}$ in $L_2$. Empirically, our extensive experiments on the simulated and real world high dimensional datasets justify that a pre-trained transformer can successfully perform PCA by simultaneously estimating the first $k$ eigenvectors and eigenvalues. These findings demonstrate that transformers can efficiently extract low dimensional patterns from high dimensional data, shedding light on the potential benefits of using pre-trained LLM to perform inference on high dimensional data.

## 1 INTRODUCTION

Large Language Models (LLMs) have demonstrated significant success in learning and performing inference on real world high dimensional datasets. Most modern LLMs use transformers (Vaswani, 2017) as their backbones. Transformers are a class of models that demonstrate significant advantages over the previous neural network models using recurrent architectures, achieving many state-of-the-art performance in learning tasks including natural language processing (Wolf et al., 2020) and computer vision (Khan et al., 2022). However, the underlying mechanism for the success of transformers remains largely a mystery to researchers. One critical and most fundamental questions is why this model works well when adapting to large volume of data used in the training of LLMs.

It is well known in the machine learning and statistics community that high dimensional inference are subject to *the Curse of Dimensionality*. Hence, practioners use various of methods to perform dimension reduction in the covariate space before they perform subsequent inferential tasks. However, performing inference using pre-trained LLMs does not require researchers to perform dimension reduction manually (Ma et al., 2023). Instead, LLMs are able to extract the essential information from the input texts. This work attempts to understand how transformers are able to extract low dimensional patterns from large volume of data via answering the question raised in the title. We perform our study through theoretically analyzing the transformer model to show its approximation and generalization capacities in performing the PCA task.

PCA is one of the most important dimension reduction methods used in practice. In particular, it has fundamental utilities in machine learning (Bishop & Nasrabadi, 2006; Hastie et al., 2009), high dimensional statistics (Fan et al., 2020; Wainwright, 2019), and econometrics (Bai et al., 2008). For the most part, PCA is a sub-procedure for algorithms solving more complicated problems. Therefore, understanding how well transformers perform PCA is also of great importance to find potential new utilities where LLMs succeed in.

Although PCA is a standard unsupervised learning task that does not require a particular machine learning model, we show that pretraining a transformer can equip it with the capacity of performing PCA on unseen instances. The pretraining step studied in this work follows a supervised learning paradigm where the input is the covariate matrix and the label is designed to be its top $k$ eigenvectors. Some existing works also make attempt to study the universal approximation power of transformers on classes of functions including (Pérez et al., 2021; Wei et al., 2022; Yun et al., 2019). However, the

mapping from matrix to eigenvectors is not easily integrated into their framework. Moreover, their work does not imply the results obtained here as we provide approximation errors for the principle eigenvectors and the corresponding generalization error bounds given by the pretraining procedure.

**Contributions.**   We summarize our major contributions as follows:

1. We rigorously show that a pre-trained transformer can perform PCA and give approximation error bound. The proof is constructive where we utilize the logical similarities between the forward propagation on transformers and the Power Method to bound the approximation error;

2. We further provide upper bounds on the generalization error for the empirical risk minimizer in the pre-training task. Coupling with the approximation error and making tradeoff between the different terms, we show that transformers can generalize with $L_2$ error rate as fast as $n^{-1/5}$ with high probability where $n$ is the number of pre-trained samples;

3. We systematically evaluate the performance of PCA with different parameter value combinations. These empirical results demonstrate that transformers can perform very well in extracting principal eigenvectors and eigenvalues from data, even in regions where theoretical results are hard to obtain, given a proper pretraining procedure.

## 1.1 RELATED WORKS

This work is related to a few different branches in the literature.

**In Context Learning of Transformers.**   Some recent works studied the in-context learning (ICL) capacities of Transformers (Garg et al., 2022; Bai et al., 2024). In particular, (Bai et al., 2024) considered the approximation and generalization properties of transformers on the ICL tasks, including many linear regression and logistic regression setups. The problem of PCA is a standard unsupervised learning problem. Hence, it differs from ICL in that there is no individual label that the model needs to learn. Akyürek et al. (2022); Von Oswald et al. (2023) considered the approximation of transformers on gradient descent when performing ICL. In this work, the proof machine utilizes the Power Method. We also notice that performing gradient descent is difficult to obtain the eigenvectors as no explicit functional form is given. To the best of authors' knowledge, this is the first work that provides theoretical guarantees for the transformers' approximation of the Power method in the literature. Other related works on the more practical side of ICL can be found in Dong et al. (2022) and reference therein.

**Other Theoretical Works on Transformers.**   Many other attempts are made to theoretically understand transformers. Yun et al. (2019) studied the universal approximation properties of transformers on sequence-to-sequence functions. Pérez et al. (2021); Bhattamishra et al. (2020); Liu et al. (2022) studied the computational power of transformers. Hron et al. (2020) studied the limit of infinite width multi/single head attentions. Yao et al. (2021) showed that transformers can process bounded hierarchical languages and demonstrate better space complexity than the recurrent neural networks.

**Notations**   In this work we follow the following notation conventions. The vector valued variable is given by boldfaced characters. We denote $[n] := \{1, \ldots, n\}$ and $[i : j] := \{i, i+1, \ldots, j\}$ for $i < j$. The universal constants are given by $C$ and is ad hoc. For a vector $\boldsymbol{v}$ we denote $\|\boldsymbol{v}\|_2$ as its $L_2$ norm. For a matrix $\boldsymbol{A} \in \mathbb{R}^{m \times n}$ we denote its operator norm as $\|\boldsymbol{A}\|_2 := \sup_{\boldsymbol{v} \in \mathbb{S}^{n-1}} \|\boldsymbol{A}\boldsymbol{v}\|_2$. Given two sequences $a_n$ and $b_n$, we denote $a_n \lesssim b_n$ or $a_n = O(b_n)$ if $\limsup_{n \to \infty} |\frac{a_n}{b_n}| < \infty$ and $a_n = o(b_n)$ if $\limsup_{n \to \infty} |\frac{a_n}{b_n}| = 0$.

**Organizations**   The rest of the paper is organized as follows: Section 2 describes the learning problems, the idea of our constructive proof, and reviews standard contexts; Section 3 provides rigorous theoretical results; Section 4 provides extensive experimental details and results; Section 5 discusses the limitations and potential future works. The detailed proofs and additional figures in experiments are delayed to the appendix. The supplementary materials include the code for the experiments.

## 2 APPROXIMATE PCA BY PRE-TRAINED TRANSFORMERS

This section discusses how we construct a multi-layered transformer model such that forward propagate along the it gives us the left principle eigenvectors of the input matrix. Our discussions is splitted into 3 subsections: In 2.1 we review the mathematical forms of the Transformer model in this work; In 2.2 we review the classical power method algorithm to perform PCA and connects it with the multiphase Transformer design; In 2.3 we demonstrate how we perform supervised pre-training to achieve a model in 2.2.

### 2.1 THE TRANSFORMERS

We consider the context learning problem on the transformer model. Under this formulation, we have the following definition regarding an attention layer. These definitions are similar to that given by Bai et al. (2024).

**Definition 1** (Attention Layer). *A self-attention layer with $M$ heads is denoted as $Attn_{\boldsymbol{\theta}_1}(\cdot)$ with parameters $\boldsymbol{\theta}_1 = \{(\boldsymbol{V}_m, \boldsymbol{Q}_m, \boldsymbol{K}_m)\}_{m \in [M]} \subset \mathbb{R}^{D \times D}$. On input sequence $\boldsymbol{H} \in \mathbb{R}^{D \times N}$,*

$$Attn_{\boldsymbol{\theta}_1}(\boldsymbol{H}) = \boldsymbol{H} + \frac{1}{N} \sum_{m=1}^{M} (\boldsymbol{V}_m \boldsymbol{H}) \sigma \left( (\boldsymbol{Q}_m \boldsymbol{H})^\top (\boldsymbol{K}_m \boldsymbol{H}) \right),$$

*where $\sigma$ is the relu activation function.*

**Remark 1.** *Note that, instead of the concatenated feature given by multi-head attention, we consider simple average on the multi-head output. And the activation function we considered is Relu instead of Softmax that appears in most empirical works. (Shen et al., 2023) empirically verified that Relu transformers are strong alternatives to Softmax transformers. We also omit the layer-wise normalization used to stablize the training procedure. These adaptation are designed for the technical convinience. In the simulation section we carefully evaluate the effect of these additional features.*

The following two layers defines the classical MLP layers with residual connections.

**Definition 2** (MLP Layer). *A MLP layer with hidden dimension $D'$ is denoted as $MLP_{\boldsymbol{\theta}}(\cdot)$ with parameter $\boldsymbol{\theta}_2 \in (\boldsymbol{W}_1, \boldsymbol{W}_2) \in \mathbb{R}^{D' \times D} \times \mathbb{R}^{D \times D'}$. On any input sequence $\boldsymbol{H} \in \mathbb{R}^{D \times N}$, we define*

$$MLP_{\boldsymbol{\theta}_2}(\boldsymbol{H}) := \boldsymbol{H} + \boldsymbol{W}_2 \sigma(\boldsymbol{W}_1 \boldsymbol{H}).$$

Then we use the above two definitions on the MLP and the Attention layers to define the Transformers.

**Definition 3** (Transformer). *We define a transformer $TF_{\boldsymbol{\theta}}(\cdot)$ as a composition of self-attention layers with MLP layers. Consider output dimension to be $\tilde{D}$, the . In particular, a L-layered Transformer is defined by*

$$TF_{\boldsymbol{\theta}}(\boldsymbol{H}) := \tilde{\boldsymbol{W}}_0 \times MLP_{\boldsymbol{\theta}_2^L}(Attn_{\boldsymbol{\theta}_1^L}(\cdots MLP_{\boldsymbol{\theta}_2^1}(Attn_{\boldsymbol{\theta}_1^1}(\boldsymbol{H}))) \times \tilde{\boldsymbol{W}}_1,$$

*where $\tilde{\boldsymbol{W}}_0 \in \mathbb{R}^{d_1 \times D}$ and $\tilde{\boldsymbol{W}}_1 \in \mathbb{R}^{N \times d_2}$.*

We use $\boldsymbol{\theta}$ to denote all the parameters in the transformer and the super-index $\ell$ to denote the parameter matrix corresponds to the $\ell$-th layer. Under such definition, the parameter $\boldsymbol{\theta}$ is given by

$$\boldsymbol{\theta} = \{\{(\{\boldsymbol{Q}_m^\ell, \boldsymbol{K}_m^\ell, \boldsymbol{V}_m^\ell\}_{m \in [M]}, \boldsymbol{W}_1^\ell, \boldsymbol{W}_2^\ell)\}_{\ell \in [L]}, \tilde{\boldsymbol{W}}_0, \tilde{\boldsymbol{W}}_1\}.$$

**Remark 2.** *The model's Lipchitzness can be strictly governed by the following aspects: (1) The number of layers; (2) The number of heads; (3) The maximum operator norm of the parameters. These results further lead to an upper bound on the generalization error. Collecting the above three aspects, we define the following operator norm of the parameters*

$$\|\boldsymbol{\theta}\|_{op} := \max_{\ell \in [L]} \left\{ \max_{m \in [M^\ell]} \left\{ \|\boldsymbol{Q}_m^\ell\|_2, \|\boldsymbol{K}_m^\ell\|_2 \right\} + \sum_{m=1}^{M^\ell} \|\boldsymbol{V}_m^\ell\|_2 + \|\boldsymbol{W}_1^\ell\|_2 + \|\boldsymbol{W}_2^\ell\|_2 \right\},$$

*where $M^\ell$ is the number of heads of the $\ell$-th attention layer. It is shown in (Bai et al., 2024) that such norm relates to the Lipschitz constant of transformers.*

The two additional matrices $\tilde{\boldsymbol{W}}_0$ and $\tilde{\boldsymbol{W}}_1$ serve for the dimension adjustment purpose such that the output of $TF_{\boldsymbol{\theta}}()$ will be of dimension $\mathbb{R}^{d_1 \times d_2}$.

## 2.2 THE POWER METHOD

The power method is an efficient iterative algorithm to solve for the principle eigenvectors and eigenvalues of an asymmetric matrix (Golub & Van Loan, 2013). The formal statement is given by algorithm 1.

---

**Algorithm 1:** Power Method for the Left Singular Vectors

---

**Data:** Matrix $\boldsymbol{X} \in \mathbb{R}^{d \times N}$, Number of Iterations $\tau$

Symmertize $A = XX^\top \in \mathbb{R}^{d \times d}$ ;

Let the set of eigenvectors be $\mathcal{V} = \{\}$. Initialize $A_1 \leftarrow A$;

**for** $\ell \leftarrow 1$ *to* $k$ **do**

    Sample a random vector $\boldsymbol{v}_{0,\ell} \in \mathbb{S}^{N-1}$. Initialize $\boldsymbol{v}_\ell^{(0)} \leftarrow \boldsymbol{v}_{0,\ell}$;

    **for** $t \leftarrow 1$ *to* $\tau$ **do**

        Apply the procedure to obtain the principle eigenvector $\boldsymbol{v}_\ell^{(t)} = \frac{A_\ell \boldsymbol{v}_\ell^{(t-1)}}{\|A_\ell \boldsymbol{v}_\ell^{(t-1)}\|_2}$;

    Let $\mathcal{V} \leftarrow \mathcal{V} \cup \{\boldsymbol{v}_\ell^{(\tau)}\}$;

    Compute the eigenvalue estimate $\widehat{\lambda}_\ell \leftarrow \|A_\ell \boldsymbol{v}_\ell^{(\tau)}\|_2$;

    Update the matrix by $A_{\ell+1} = A_\ell - \widehat{\lambda}_\ell \boldsymbol{v}_\ell^{(\tau)} \boldsymbol{v}_\ell^{(\tau),\top}$ ;

**return** $\mathcal{V}$;

---

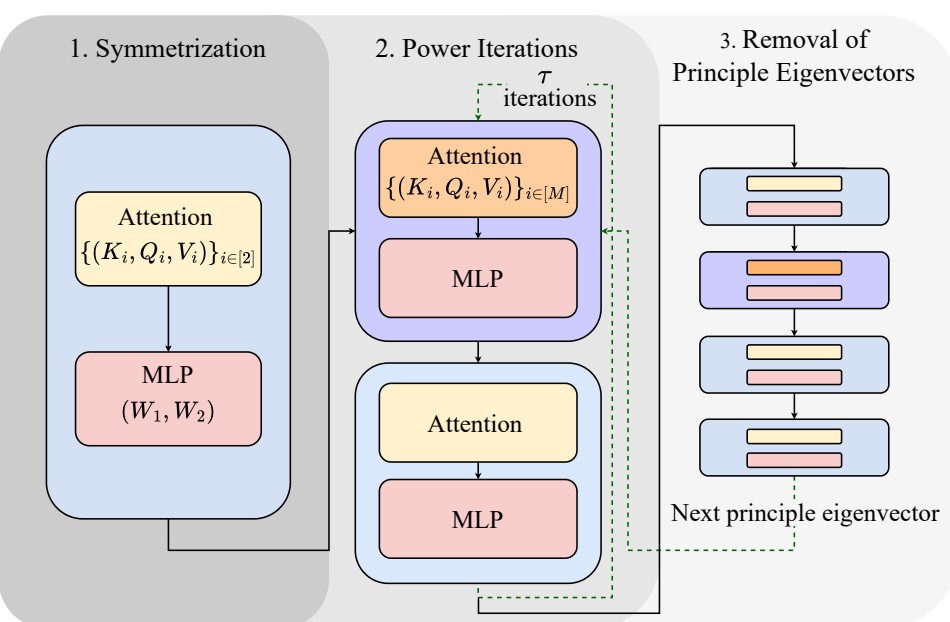

Figure 1: **The Approximation by Transformers.** The above diagram illustrates the design of our transformer model. There are three important sub-networks in the design: *(1) The Symmetrization* sub-network symmetrizes $\boldsymbol{X}$ and stamp $\boldsymbol{X}\boldsymbol{X}^\top$ in the output, which corresponds to the first step in the Power Method. *(2) The Power Iterations* sub-network performs in total of $\tau$ iterations for each of the principle eigenvectors, corresponds to the iterative update step in the Power Method. *(3) The Removal of Principle Eigenvectors* subnetwork performs the estimate of $\widehat{\lambda}_\ell$ and the update of matrix $A_\ell$ in the Power Method. Finally, we apply $\tilde{\boldsymbol{W}}_0$ and $\tilde{\boldsymbol{W}}_1$ to adjust the dimension of the output. The different colors in the diagram corresponds to the different type of layers: *(1) Yellow Blocks* denote the Attention layer with 2 heads. *(2) Orange Blocks* denote the multihead transformers with larger $M \gg 2$. *(3) Pink Blocks* denote the MLP layer.

The algorithm itself first generate the symmetrized covariate matrix $A = XX^\top$. Then, for each eigenvector that we hope to recover, the power method generates a random vector uniformly distributed on the unit sphere. Then we iterately take the matrix product between the symmetric matrix $A$ and $\boldsymbol{v}_{0,\ell}$ followed by normalizing it. Note that such iterative matrix product can finally converge to the top-1 principle eigenvector as the iterations go further.

The iterative structure is akin to the forward propagation of the transformer achitecture. Given by such similarities, we show that there exists a parameter setup for Transformers that the forward propogation performs principle component analysis. The challenge is in constructing parameters that approximate the complete procedure of the power method.

In figure 1, we provide an *approximate* algorithm for the power method through the lens of a forward propagation on the transformer. This algorithm dissects the approximation of power method into the propagation along multiple sub-networks, each phase corresponds to a single step in the power method. Combining them, we show in section 3 that Transformers achieve good approximation guarantees.

### 2.3 Pretraining via Supervised Learning

The standard PCA problem is unsupervised where no labels given. However, the Transformers are usually used in the supervised learning setup. To make full use of Transformers in the PCA task, we need to perform supervised pre-training. In our theoretical analysis, we construct the input of the Transformer as a *context-augmented matrix* given by the following

$$\boldsymbol{H} = \begin{bmatrix} \boldsymbol{X} \\ \boldsymbol{P} \end{bmatrix} \in \mathbb{R}^{D \times N}, \qquad \boldsymbol{P} = \begin{bmatrix} \tilde{\boldsymbol{p}}_{1,1}, \ldots, \tilde{\boldsymbol{p}}_{1,N} \\ \tilde{\boldsymbol{p}}_{2,1}, \ldots, \tilde{\boldsymbol{p}}_{2,N} \\ \vdots \\ \tilde{\boldsymbol{p}}_{\ell,1}, \ldots, \tilde{\boldsymbol{p}}_{\ell,N} \end{bmatrix} \in \mathbb{R}^{(D-d) \times N},$$

where the matrix $\boldsymbol{P}$ contains contextual information, which is specified in section 3. The design also makes sure $\boldsymbol{P}$ is unrelated to $\boldsymbol{X}$. In the experiments, we show that the auxillary matrix $\boldsymbol{P}$ is not necessary for the pre-trained Transformer to perform PCA with high accuracy. For the output, our theoretical analysis gives the following matrix

$$TF_{\boldsymbol{\theta}}(\boldsymbol{H}) = \begin{bmatrix} \widehat{\boldsymbol{v}}_1^\top & \ldots & \widehat{\boldsymbol{v}}_k^\top \end{bmatrix}^\top \in \mathbb{R}^{dk}$$

which corresponds to the estimated principle eigenvectors of the matrix $\boldsymbol{X}$.

**The Learning Problem.** Consider a set of samples $\{\boldsymbol{X}_i\}_{i \in [u]}$ i.i.d. sampled from some distribution $p_{\boldsymbol{X}}$, we construct their oracle top-$k$ principle components as $\boldsymbol{V}_i = \begin{bmatrix} \boldsymbol{v}_1^{i,\top} & \ldots & \boldsymbol{v}_k^{i,\top} \end{bmatrix}^\top$ and the context-augmented input matrix as $\boldsymbol{H}_i$ for each $\boldsymbol{X}_i$. Then, the pretraining procedure is given by minimizing the following objective for some convex loss function $L(\cdot, \cdot) : \mathbb{R}^{dk} \times \mathbb{R}^{dk} \to \mathbb{R}$,

$$\widehat{\boldsymbol{\theta}} = \underset{\boldsymbol{\theta} \in \Theta(B_{\boldsymbol{\theta}}, B_M)}{\arg\min} \sum_{i=1}^{u} L(TF_{\boldsymbol{\theta}}(\boldsymbol{H}_i), \boldsymbol{V}_i). \tag{1}$$

Here we consider $\Theta(B_{\boldsymbol{\theta}}) := \{\boldsymbol{\theta} : \|\boldsymbol{\theta}\| \le B_{\boldsymbol{\theta}}, \max_\ell M^\ell \le B_M\}$ to be the space of parameters. We also consider guarantees in the $L_2$ norm which states that $L(\boldsymbol{x}_1, \boldsymbol{x}_2) := \|\boldsymbol{x}_1 - \boldsymbol{x}_2\|_2$ in the theoretical part. Since $\widehat{\boldsymbol{\theta}}$ given by minimizing the empirical risk is not obtainable in practice, our theory only gives guarantee on the empirical risk minimizer. We further show that the local minimizers obtained through stochastic gradient optimization achieve good empirical performance in section 4.

### 3 Theoretical Results

This section presents our theoretical results and the idea of taking each steps in the proof. Our proof constructs a particular instance of the transformers and show that the forward propagation on our constructed instance approximates the Power Method. We also carefully design the contextual matrix $\boldsymbol{P}$, explained as follows.

**The Design of Auxillary Matrix.** Our design of the matrix $\boldsymbol{P}$ consists of three parts:

1. *Place Holder.* For $\ell \in \{1\} \cup [4 : k+3]$ and $i \in [N]$, we let $\tilde{\boldsymbol{p}}_{\ell,i} = \boldsymbol{0} \in \mathbb{R}^{d \times 1}$. The place holders in $\boldsymbol{P}$ records the intermediate results in the forward propagation.

2. *Identity Matrix.* We let $\begin{bmatrix} \tilde{\boldsymbol{p}}_{2,1} & \dots & \tilde{\boldsymbol{p}}_{2,N} \end{bmatrix} = \begin{bmatrix} \boldsymbol{I}_d & \boldsymbol{0}_{d \times (N-d)} \end{bmatrix}$. The identity matrix in $\boldsymbol{P}$ helps us screen out all the covariates $\boldsymbol{X}$ in the forward propagation.

3. *Random Samples on the Hypersphere.* We let $\tilde{\boldsymbol{p}}_{3,1}, \dots \tilde{\boldsymbol{p}}_{3,k}$ be the i.i.d. samples uniformly distributed on $\mathbb{S}^{d-1}$. The random samples on the sphere corresponds to the initial vectors $\boldsymbol{v}_{0,\ell}$ for $\ell \in [k]$ in algorithm 1.

Given the above construction on the auxillary matrix $\boldsymbol{P}$, we are ready to state the existence theorem in this work, given as follows.

**Theorem 3.1** (Transformer Approximation of the Power Iteration). *Denote the eigenvalues of $\boldsymbol{X}\boldsymbol{X}^\top$ to be $\lambda_1 > \lambda_2 > \dots > \lambda_k > \dots$. Let $\Delta := \min_{1 \le i < j \le k} |\lambda_i - \lambda_j|$. Assume that the eigenvalues of $\boldsymbol{X}$ satisfy $\|\boldsymbol{X}\|_2 \le B_X$. Assume that the initialized vectors $\tilde{\boldsymbol{p}}_{3,1}, \dots \tilde{\boldsymbol{p}}_{3,N}$ satisfy $\tilde{\boldsymbol{p}}_{3,i}^\top \boldsymbol{v}_i \ge \delta$ for all $i \in [k]$ and make the rest of the vectors $\boldsymbol{0}$. Then, there exists a transformer model with number of layers $L = 2\tau + 4k + 1$ and number of heads $M \le \lambda_1^d \frac{C}{\epsilon^2}$ with $\tau \le \frac{\log(1/\epsilon_0\delta)}{\epsilon_0}$ such that for all $\epsilon_0, \epsilon > 0$, the final output $\widehat{\boldsymbol{v}}_1, \dots, \widehat{\boldsymbol{v}}_k$ given by the transformer model achieve*

$$\|\widehat{\boldsymbol{v}}_{\eta+1} - \boldsymbol{v}_{\eta+1}\|_2 \le C\tau\epsilon\lambda_1^2 + \frac{C\lambda_1\sqrt{\epsilon_0}}{\Delta}\prod_{i=1}^{\eta}\frac{5\lambda_{i+1}}{\Delta}.$$

*Moreover, consider the accuracy of multiple $\boldsymbol{v}$s as a whole. There exists $\boldsymbol{\theta}$ such that*

$$L\left(TF_{\widehat{\boldsymbol{\theta}}}(\boldsymbol{H}), \boldsymbol{V}\right) \le C\tau\epsilon k\lambda_1^2 + C\left(\frac{\epsilon_0\lambda_1^2}{\Delta^2}\sum_{\eta=1}^{k-1}\prod_{i=1}^{\eta}\frac{25\lambda_{i+1}^2}{\Delta^2}\right)^{1/2}.$$

**Remark 3.** *The approximation error consists of two terms. The frist term comes from the approximation of the Power Method iterations by transformers. The second term comes from the error caused by finite iteration $\tau$. To acquire a more direct account of the error terms and its order of magnitude, we consider a special case where the eigenvalues $\lambda_1 \asymp \lambda_2 \asymp \dots \asymp \lambda_k \asymp \Delta$. Then our results boils down to*

$$\left\| TF_{\boldsymbol{\theta}}(\boldsymbol{H}) - \begin{bmatrix} \boldsymbol{v}_1^\top, \boldsymbol{v}_2^\top, \dots, \boldsymbol{v}_k^\top \end{bmatrix}^\top \right\|_2 \le C\tau\epsilon k\lambda_1^2 + C\frac{\lambda_1}{\Delta}\sqrt{k\epsilon_0}.$$

*These results hide dimension $d$ in the universal constant. Hence the dimension significantly affects the approximation properties of transformers. Our experimental results in section 4 also indicate that learning high dimensional principle eigenvectors is challenging.*

We show that the conditions on $\tilde{\boldsymbol{p}}_{3,1}, \dots, \tilde{\boldsymbol{p}}_{3,N}$ can be achieved through sampling from isotorpic Gaussians, given by the following lemma.

**Lemma 3.1.** *Let $\boldsymbol{y} \in \mathbb{R}^d$ be a random vector with isotropic Gaussian as its probability density. Consider $\boldsymbol{x} = \frac{\boldsymbol{y}}{\|\boldsymbol{y}\|_2}$. Let $\boldsymbol{v}$ be any unit length vector, then we have for all $\delta < \frac{1}{2}d^{-1}$, $\mathbb{P}\left(|\boldsymbol{v}^\top \boldsymbol{x}| \le \delta\right) \le \frac{1}{\sqrt{\pi}}\sqrt{\delta} + \exp\left(-C\delta^{-\frac{1}{2}}\right)$. Therefore, for all $\delta < \frac{1}{2}d^{-1}$, the event in theorem 3.1 is achieved with*

$$\mathbb{P}\left(\exists i \in [k] \text{ such that } \boldsymbol{x}_i^\top \boldsymbol{v}_i \le \frac{\delta}{\sqrt{d}}\right) \le \frac{k\sqrt{\delta}}{\sqrt{\pi}} + k\exp(-C\delta^{-1}).$$

Given the approximation error provided by theorem 3.1, we further provide the generalization error bound for the ERM defined by equation 1. This requires us to consider the following regularity conditions on the underlying distribution of $\boldsymbol{X}\boldsymbol{X}^\top$ (which also translates to the distribution of $\boldsymbol{X}$).

**Assumption 1.** *The distribution of $\boldsymbol{X}\boldsymbol{X}^\top$ supports on*

$$\mathbb{X} := \left\{ A : A \in \boldsymbol{S}_{++}^d, B_X \ge \lambda_1(A) > \lambda_2(A) > \dots > \lambda_k(A), \inf_{1 \le i < j \le k} \lambda_i(A) - \lambda_j(A) \ge \Delta \right\}.$$

**Remark 4.** *The above assumption can be easily generalized to distribution that supports on $\mathbb{X}$ with high probability. Examples of such distribution include the Wishart distribution under the Gaussian design. In this work, we stick to the simplest case where the maximum eigenvalue is bounded from above.*

Given the above assumption, we are ready to state the generalization bound.

**Proposition 1.** *With probablity at least $1 - \xi$, the ERM solution $\widehat{\boldsymbol{\theta}}$ satisfies*

$$\mathbb{E}\left[L\left(TF_{\widehat{\boldsymbol{\theta}}}(\boldsymbol{H}), \boldsymbol{V}\right) | \widehat{\boldsymbol{\theta}}\right] \leq \inf_{\boldsymbol{\theta} \in \Theta(B_{\boldsymbol{\theta}}, B_M)} \mathbb{E}\left[L\left(TF_{\boldsymbol{\theta}}(\boldsymbol{H}), \boldsymbol{V}\right)\right]$$
$$+ C\sqrt{\frac{k^3 L B_M d^2 \log(B_{\boldsymbol{\theta}} + B_X + k) + \log(1/\xi)}{n}}.$$

Together with the bound given by theorem 3.1 and lemma 3.1, which essentially give a high probability upper bound on $\inf_{\boldsymbol{\theta} \in \Theta(B_{\boldsymbol{\theta}}, B_M)}$ we can derive a general upper bound on the generalization error, given as follows.

**Corollary 3.1.1.** *Under assumption 1, with probability at least $1 - \xi - \frac{k\sqrt{\delta}}{\sqrt{\pi}} - k \exp\left(-C\delta^{-1/2}\right)$ for all $\delta < d^{-1}$ we have for all $\epsilon, \epsilon_0 > 0$,*

$$\mathbb{E}\left[L\left(TF_{\widehat{\boldsymbol{\theta}}}(\boldsymbol{H}), \boldsymbol{V}\right) | \widehat{\boldsymbol{\theta}}\right] \leq \mathbb{E}\left[C\tau\epsilon k\lambda_1^2 + C\left(\frac{\epsilon_0 \lambda_1^2}{\Delta^2} \sum_{\eta=1}^{k-1} \prod_{i=1}^{\eta} \frac{25\lambda_{i+1}^2}{\Delta^2}\right)^{1/2}\right]$$
$$+ C\sqrt{\frac{k^3 \log(\delta/\epsilon_0)\lambda_1^d d^2 \log(B_{\boldsymbol{\theta}} + B_X + k) + \log(1/\xi)}{n\epsilon_0 \epsilon^2}}.$$

**Remark 5.** *If we consider optimizing the bound w.r.t. $\epsilon_0$ and $\epsilon$, we obtain that $\mathbb{E}\left[L(TF_{\widehat{\boldsymbol{\theta}}}(\boldsymbol{H}), \boldsymbol{V})|\widehat{\boldsymbol{\theta}}\right] \lesssim n^{-1/5}$ given that the rest of the parameters are of constant scales. It is not known if the results are improvable or not and the authors believe this question worth future explorations.*

## 4 SIMULATIONS

In this section, we verify the theoretical result in section 3 on synthetic and real-world datasets. Our experiments include both prediction of eigenvalues and eigenvectors. For synthetic datasets, we generate samples according to normal distributions. We focus on evaluating the effects of three major parameters: **(1)** The Impact of $D$; **(2)** The Impact of Number of Layers; **(3)** The Impact of $k_{\text{train}}$[1]. For real-world datasets, we perform experiments on MNIST (LeCun et al., 1998) and Fashion-MNIST (Xiao et al., 2017). *All the results presented in this section are errors on the testing set.*

**Data Preparation.** For synthetic data $\boldsymbol{X} \in \mathbb{R}^{D \times N}$, we generate each column with a randomly initialized multivariate Gaussian distribution $\mathcal{N}(\mu, \Sigma)$. We then generate the labels as the top-$k$ eigenvalues $\lambda$ and eigenvectors $V$ of the empirical covariance matrix $\boldsymbol{X}^\top \boldsymbol{X}/(N-1)$ via `numpy.linalg.eigh`. For real world dataset, we apply SVD to reduce the dimensionality of both datasets and evaluate whether the transformer, previously trained on multivariate Gaussian data, are capable of performing PCA on those real-world datasets. If the transformer successfully learns to perform PCA, we expect comparable performance on both synthetic and real-world data. For more details on data generation and configuration, please refer to table 2 in appendix C.2.

**Model.** We use the GPT2-architecture transformer (Radford et al., 2019) as our backbone model. We follow most settings in Garg et al. (2022), but replace the Softmax attention with ReLU attention as constructed in definition 1. We also provide a empirical comparison between Softmax and ReLU attention in Figure **??** in the Appendix. We use a slighly differernt architectures to predict eigenvalues and eigenvectors. For eigenvalues prediction, we flatten the transformer output

---

[1] We denote $k_{\text{train}}$ as the value of $k$ used in training

Table 1: **Cosine Similarity for Different $k$.** We dentoe $k_{\text{train}}$ as the number of eigenvectors to predict during training. For example, for $k_{\text{train}} = 4$, the model is trained to predict 4 eigenvectors.

| k-th eigenvec. | k=1 | k=2 | k=3 | k=4 |
|---|---|---|---|---|
| $k_{\text{train}} = 4$ | 0.891(0.006) | 0.616(0.038) | 0.282(0.047) | 0.120(0.022) |
| $k_{\text{train}} = 3$ | 0.908(0.011) | 0.706(0.023) | 0.366(0.018) | - |
| $k_{\text{train}} = 2$ | 0.903(0.006) | 0.647(0.019) | - | - |
| $k_{\text{train}} = 1$ | 0.894(0.009) | - | - | - |

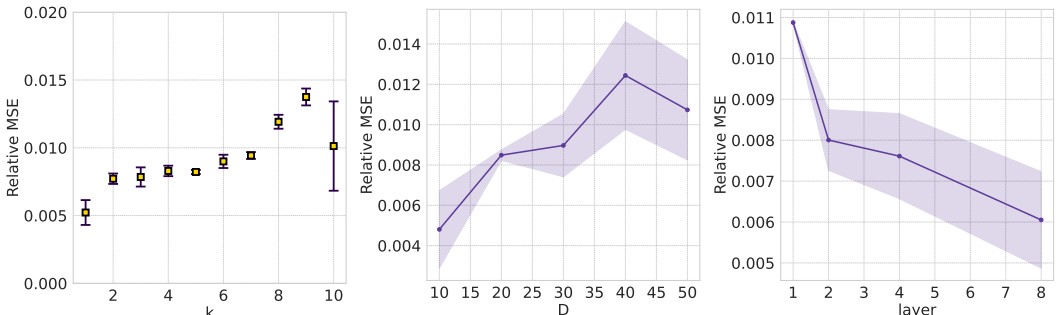

Figure 2: **Comparisons of Eigenvalue Prediction on Synthetic Data.** *(1) Left: Evidence of Transformer's Ability to Predict Multiple Eigenvalues.* We use a small transformer (layer = 3, head = 2, embedding = 64) to predict top 10 eigenvalues with $D = 20$ and $N = 50$. All of the (Relative MSE) of 10 eigenvalues are below 2%, verifying that the transformer can predict eigenvalue very well. Additionally, the error of prediction grows slightly with $k$. We note that: (i) Higher-order eigenvalues require additional iterations and models with more layers. (ii) Smaller eigenvalues are more sensitive to the fluctuations in the predicted values under the relative MSE metric. *(2) Middle: Predictions of eigenvalues with different input dimension $D$.* We use a small transformer and use $N = 10$ in this experiment. We show that prediction errors increase significantly as dimension scales up, corroborating our theoretical remark 3. *(3) Right: Predictions of eigenvalues with different number of layers.* We use the same input as the previous multiple eigenvalues predictions experiment, and use a small transformer to predict top-3 eigenvalues. As the number of layers grow, the model performs better on eigenvalue prediction, which aligns with the result in theorem 3.1.

$TF_{\boldsymbol{\theta}}(\boldsymbol{H}) \in \mathbb{R}^{N \times D}$ and use a linear layer $W_\lambda \in \mathbb{R}^{(N \cdot D) \times k}$ to readout the top $k$ eigenvalues. As for eigenvectors, we use one more linear layer $W_v \in \mathbb{R}^{(N \cdot D) \times (k \cdot D)}$ to readout $k$ eigenvectors concatenated in a 1-dimension vector. We use a transformer with layer = 3, head = 2, and embedding size = 64 to speed up the training process for most settings and find that it is sufficient to predict multiple eigenvalues and top-1 eigenvector well, see below sections for detailed discussion.

**Metrics.** For eigenvalues, we use relative mean squared error (RMSE) as loss function $\mathcal{L}_{\text{RMSE}}$ and evaluation metric. For the loss of predicting top-$K$ eigenvalue, the loss function is defined as following

$$\mathcal{L}_{\text{RMSE}}(\lambda_i, \hat{\lambda}_i) := \frac{1}{K} \sum_{i=1}^{K} \frac{\lambda_i - \hat{\lambda}_i}{\lambda_i + \epsilon}.$$

For eigenvectors, we use cosine similarity as loss function and evaluation metric. For predicting $k$ eigenvectors, the loss function is defined as

$$\mathcal{L}_{\cos}(v_i, \hat{v}_i) := \frac{1}{K} \sum_{i=1}^{K} 1 - \frac{v_i \cdot \hat{v}_i}{\max(\|v_i\|_2 \cdot \|\hat{v}_i\|_2, \epsilon)},$$

where $v_i$ represent the $i$-th eigenvector. The design of these loss functions not only matches the intuition of eigenvectors and eigenvalues, but also stablize training by normalizing the loss values.

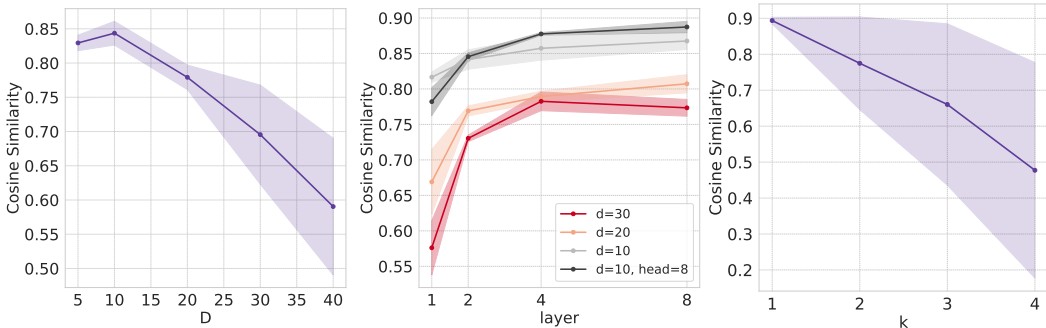

Figure 3: **Comparison of Eigenvector Prediction on Synthetic Data.** *(1) Left: Prediction of top-1 eigenvector with different input dimension $D$.* We use a small transformer and $N = 10$. As the dimension $D$ scales up the eigenvector prediction suffers significantly. *(2) Middle: Prediction of top=1 eigenvector with varying number of layers.* We start from small transformer and use $N = 5$. The result demonstrates an 'elbow effect', where we show that the increase of $L$ significantly boost the performance when $L$ is small but halt to progress for larger $L$. We believe this can be explained by the bias-variance tradeoff. *(3) Right: Predictions of eigenvectors with different numbers of $k$* We use $N = 10$ and $D = 10$ in this experiment. We use a larger transformer with layer= 12, heads= 8, and an embedding size= 256 in this experiment. The result demonstrates a decreasing prediction accuracy and increasing standard deviation as $k$ increases. We list the individual cosine similarities of the predicted k-th eigenvectors in table 1. All the evaluations in the above three figures are averaged on three runs with different random seed.

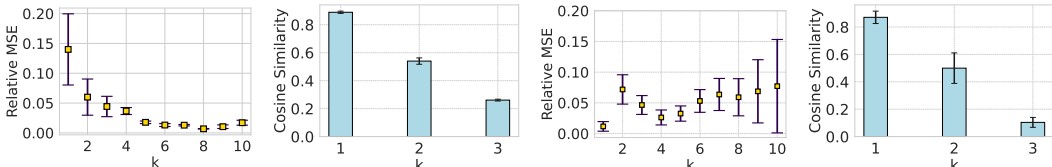

Figure 4: **Comparison of Eigenvalues and Eigenvectors Prediction on Real World Data.** *(1) Left: Predicting Top-10 Eigenvalues on MNIST (2) Second Left: Predicting Top-3 Eigenvectors on MNIST (3) Second Right: Predicting Top-10 Eigenvalues on FMNIST (4) Right: Predicting Top-3 Eigenvectors on FMNIST.* We show that on real world datasets, Transformers perform similarly to the synthetic datasets. The experimental setup for real world data is analogous to the ones performed for synthetic data.

### 4.1 SYNTHETIC DATA VS REAL WORLD DATA

**Synthetic Dataset.** The results on synthetic data are in figure 2. We first observe that transformers are capable of predicting top-10 eigenvalues with small error ($< 2\%$ error). The result also corresponds to theorem 3.1, indicating transformers are able to perform the power iteration method and generate eigenvalues with small error. For the impact of $D$, we observe the second subfigure in figure 2. In general, we discover an increasing trend of RMSE when $D$ increases, this coincides with the theoretical findings stated in remark 3. We also observe that the error of prediction slightly increases with $k$, which is natural as the prediction dimension grows larger. For the impact of layers (right subfigure of figure 2), we observe that as the number of layer increases, RMSE shows significant reduction. This matches our theoretical construction as we show the iteration of power methods correspond to the number of layers, see figure 1 for the visualization of our transformer model. One thing to highlight is higher-order eigenvalues (larger $k$) have smaller magnitudes, which are are more sensitive to fluctuations in the predicted values when using the relative MSE metric. This explains the higher variance/error of higher order eigenvalues. For eigenvectors, we also observe that transformers are capable of predicting principle eigenvectors. In particular, the cosine similarity between predicted eigenvector and ground truth is close to 1 when $D$ is small.

**Real World Dataset.** The results on real world dataset are in figure 4. We observe that transformers are also capable of predicting top-$k$ eigenvalues well on both MNIST and FMNIST. Despite the

difference in data distribution on training and test data, transformers are able to produce small error on predicting eigenvalues. Overall, we show that pretrained transformers learn PCA, and is able to generalize to other datasets as well. For eigenvectors, we can see that trained transformers show similar behavior on real world datasets when comparing to the synthetic ones. Indicating that our model actually learn to perform PCA instead of learn certain inductive bias.

## 4.2 PREDICTION WITH DIFFERENT PARAMETER COMBINATION.

**Prediction with different $D$.** The results are given in figure 3. In this experiment, we test the influence of increasing feature dimension $D$ affects the ability of a Transformer model to predict the principal eigenvector of a data matrix. We use the simplest setting with a small transformer and test on $D = 5, 10, 20, 30, 40$, $N = 10$ and predict top-1 eigenvector. As the feature dimension $D$ increases, we observe a clear trend of performance degradation of the Transformer's ability to predict the principal eigenvector accurately. This confirms our theoretical results stated in remark 3 that the feature dimension $D$ affects the approximation properties of Transformers significantly.

**Prediction with different $L$.** The results are given in figure 3. In this experiment, we change the number of layers of transformer with head $= 2$ and embedding $= 64$, and set $N = 5$ to speed up the experiment. We observe that as the number of layers increases, the testing error also decreases, but the decreasing scale is less obvious when the number of layers becomes larger. However, the rate of improvement diminishes as the number of layers becomes larger. This suggests that increasing model depth alone is not sufficient for significantly enhancing eigenvector prediction. To verify our guess, we increase the number of heads from 2 to 8 and find that the cosine similarity increases further and with a slightly steeper incline. Note that there is a sharper decrease between layer$= 1$ and layer$= 2$ across different $d$. This finding supports theorem 3.1 that we need at least 2 layers of the transformer to perform one iteration of the power method.

**Prediction with different $k$.** The results are given in table 1, and the right subfigure in figure 3 where the cosine similarity in y-axis is averaged over $k$ eigenvectors. We use $N = 10$ and $D = 10$ in this experiment. We use a larger transformer with layer$= 12$, heads$= 8$, and an embedding size$= 256$ in this experiment. As shown in figure 3, the model's ability to predict top $k$ eigenvectors decreases as more eigenvectors are predicted, with increasing standard deviation. Table 1 lists the individual cosine similarities of the predicted $k$-th eigenvectors. The results show that most errors come from high-order eigenvectors. When trained to predict $k_{\text{train}} = 4$ eigenvectors, the model performs as well at predicting the top 1 eigenvector as when trained on $k_{\text{train}} = 1$. The result shows that most prediction errors come from high-order eigenvectors. This suggests that the pivotal difficulty is in the prediction of higher order eigenvectors.

## 5 DISCUSSIONS

This section discusses the limitations in this work and potential future working directions.

**Limitations.** Our limitations in the theoretical results can be summarized as follows: **(1)** From the theoretical perspective, our results guarantee the performance of ERM solutions whereas the true estimator is obtained through stochastic gradient descent method; **(2)** Our theoretical results utilize the context-augmented matrix $P$, which is verified removable from our empirical results. It is conjectured that this is also not necessary in theory.

**Future Works.** Beyond resolving the limitations in this work, other future working directions from this work include: **(1)** Extend the results for relu transformers to softmax transformers. This step requires researchers to develop a new approximation bound for the softmax function; **(2)** Certify whether the rate $n^{-1/5}$ is sharp or not. The authors believe that this rate is improvable but it remains quite challenging; **(3)** The Spectral Method. Many mordern high dimensional statistical questions can be resolved using the spectral method, which relies on the PCA as a sub-procedure. It is of general interest to see if these problems can be solved similarly by Transformers.

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

# A  ADDITIONAL THEORETICAL BACKGROUND

**Definition 4** (Sufficiently Smooth $d$-variate function). *Denote* $\mathsf{B}^d_\infty(R) := [-R, R]^d$ *as the standard* $\ell_\infty$ *ball in* $\mathbb{R}^d$. *We say a function* $g : \mathbb{R}^d \to \mathbb{R}$ *is* $(R, C_\ell)$ *smooth if for* $s = \lceil (d-1)/2 \rceil + 2$, $g$ *is a* $C^s$ *function on* $\mathsf{B}^d_\infty(\mathbb{R})$ *and*

$$\sup_{\boldsymbol{z} \in \mathsf{B}^d_\infty(R)} \|\nabla^d g(\boldsymbol{z})\|_\infty = \sup_{\boldsymbol{z} \in \mathsf{B}^d_\infty(R)} \max_{j_1, \dots, j_i \in [d]} \left| \partial_{x_{j_1} \dots x_{j_i}} g(\boldsymbol{x}) \right| \le L_i$$

*for all* $i \in \{0, 1, \dots, s\}$, *with* $\max_{0 \le i \le s} L_i R^i \le C_\ell$.

**Definition 5** (Approximability by sum of Relus (Bai et al., 2024)). *A function* $g : \mathbb{R}^k \to \mathbb{R}$ *is* $(\epsilon_{approx}, R, M, C)$-*approximable by sum of Relus if there exists a function* $f_{M,C}$ *such that*

$$f_{M,C}(\boldsymbol{z}) = \sum_{m=1}^M c_m \sigma(\boldsymbol{a}_m^\top[\boldsymbol{z}; 1]) \text{ with } \sum_{m=1}^M |c_m| \le C, \max_{m \in [M]} \|\boldsymbol{a}_m\|_1 \le 1, \quad \boldsymbol{a}_m \in \mathbb{R}^{k+1}, c_m \in \mathbb{R},$$

*such that* $\sup_{\boldsymbol{z} \in \mathsf{B}^k_\infty(R)} |g(\boldsymbol{z}) - f_{M,C}(\boldsymbol{z})| \le \epsilon_{approx}$.

# B  PROOFS

## B.1  PROOF OF PROPOSITION 1

*Proof.* The proof follows from (Wainwright, 2019), using the fact that for all $\boldsymbol{\theta} \in \Theta(B_{\boldsymbol{\theta}}, B_M)$, we have

$$\frac{1}{n} \sum_{j=1}^n L\left(TF_{\widehat{\boldsymbol{\theta}}}(\boldsymbol{H}_i), \boldsymbol{V}_i\right) \le \frac{1}{n} \sum_{j=1}^n L\left(TF_{\boldsymbol{\theta}}(\boldsymbol{H}_i), \boldsymbol{V}_i\right),$$

it is not hard to show that

$$\mathbb{E}\left[L\left(TF_{\widehat{\boldsymbol{\theta}}}(\boldsymbol{H}), \boldsymbol{V}\right)\right] \le \inf_{\boldsymbol{\theta} \in \Theta(B_{\boldsymbol{\theta}}, B_M)} \mathbb{E}\left[L(TF_{\widehat{\boldsymbol{\theta}}}(\boldsymbol{H}), \boldsymbol{V})\right] + 2 \sup_{\boldsymbol{\theta} \in \Theta(B_{\boldsymbol{\theta}}, B_M)} |X_{\boldsymbol{\theta}}|,$$

where $X_{\boldsymbol{\theta}} = \frac{1}{n} \sum_{j=1}^n L(TF_{\boldsymbol{\theta}}(\boldsymbol{H}_i), \boldsymbol{V}_i) - \mathbb{E}[L(TF_{\boldsymbol{\theta}}(\boldsymbol{H}), \boldsymbol{V})]$ is the empirical process indexed by $\boldsymbol{\theta}$. The tail bound for empirical process requires us to verify a few regularity conditions (Giné & Nickl, 2016) on the function $L$ and the set $\Theta$

1. The metric entropy of an operator norm ball $\log N(\delta, B_{\|\cdot\|_{op}}(r), \|\cdot\|_{op}) \le CLB_M D^2 \log\left(1 + 2(B_\theta + B_X + k)/\delta\right)$.

2. $L(TF_{\boldsymbol{\theta}}(\boldsymbol{H}), \boldsymbol{V}) \le C\sqrt{k}$.

3. The Lipschitz condition of Transformers satisfies that for all $\boldsymbol{\theta}_1, \boldsymbol{\theta}_2 \in \Theta(B_{\boldsymbol{\theta}}, B_M)$, we have $L(TF_{\boldsymbol{\theta}_1}(\boldsymbol{H}), \boldsymbol{V}) - L(TF_{\boldsymbol{\theta}_2}(\boldsymbol{H}), \boldsymbol{V}) \le CLB_1^L \|\boldsymbol{\theta}_1 - \boldsymbol{\theta}_2\|_{op}$ where $B_1 = B_\theta^4 B_X^3$.

The first and second verifications follow immediately from J.2 in (Bai et al., 2024). The third verification is given upon noticing that as $L(\boldsymbol{x}, \boldsymbol{y}) = \|\boldsymbol{x} - \boldsymbol{y}\|_2$,

$$\sup_{\boldsymbol{\theta}, \boldsymbol{H}, \boldsymbol{V}} L(TF_{\boldsymbol{\theta}}(\boldsymbol{H}), \boldsymbol{V}) \leq C\sqrt{k}, \qquad \|\nabla_{\boldsymbol{x}} L\| \leq C.$$

Further note that $\|\tilde{\boldsymbol{W}}_0\|_2 \asymp \|\tilde{\boldsymbol{W}}_1\|_2 \asymp 1$. Given the above result, and corollary J.1 in (Bai et al., 2024), we can show that

$$L(TF_{\boldsymbol{\theta}_1}(\boldsymbol{H}), \boldsymbol{V}) - L(TF_{\boldsymbol{\theta}_2}(\boldsymbol{H}), \boldsymbol{V}) \leq CLB_1^L \|\boldsymbol{\theta}_1 - \boldsymbol{\theta}_2\|_{op},$$

where $B_1 = B_\theta^4 B_X^3$. Therefore, using the uniform concentration bound given by proposition A.4 we can show that with probability at least $1 - \xi$, we have

$$\sup_{\boldsymbol{\theta} \in \Theta(B_{\boldsymbol{\theta}}, B_M)} |X_\theta| \leq C\sqrt{k} \sqrt{\frac{LB_M D^2 \log(B_\theta + B_X + k) + \log(1/\delta)}{n}}.$$

Therefore, replacing $D$ with $Ckd$ we complete the proof. $\square$

### B.2 PROOF OF THEOREM 3.1

*Proof.* Our proof can be disected into the following setps: 1. We construct a Transformer with fixed parameters that performs (1) The computation of the symmetrized covariate matrix; (2) The approximation of the power method; (3) The removal of the principle eigenvectors; (4) Adjust the dimension of the output through multiplying the two matrices $\tilde{\boldsymbol{W}}_0$ and $\tilde{\boldsymbol{W}}_1$ on the left and right.

#### 1. The Covariate Matrix.

To compute the covariate matrix $\boldsymbol{X}\boldsymbol{X}^\top$, we construct $\boldsymbol{H} = \begin{bmatrix} \boldsymbol{X}_1, \ldots, \boldsymbol{X}_N \\ \tilde{\boldsymbol{p}}_{1,1}, \ldots, \tilde{\boldsymbol{p}}_{1,N} \\ \tilde{\boldsymbol{p}}_{2,1}, \ldots, \tilde{\boldsymbol{p}}_{2,N} \\ \vdots \\ \tilde{\boldsymbol{p}}_{\ell,1}, \ldots, \tilde{\boldsymbol{p}}_{\ell,N} \end{bmatrix} = \begin{bmatrix} \boldsymbol{X} \\ \boldsymbol{P} \end{bmatrix}$ we let

$m = 2$ and

$$\boldsymbol{V}_1^{cov} = I_D = -\boldsymbol{V}_2^{cov}, \quad \boldsymbol{Q}_1^{cov,\top} \boldsymbol{K}_1^{cov} = -\boldsymbol{Q}_2^\top \boldsymbol{K}_2 = \begin{bmatrix} \boldsymbol{0}_{N+1\times d}, I_d, \boldsymbol{0} \\ \boldsymbol{0}, \boldsymbol{0}, \boldsymbol{0} \end{bmatrix} \in \mathbb{R}^{D\times D},$$

$$\tilde{\boldsymbol{p}}_{1,\ell,j} = \boldsymbol{0}, \qquad \tilde{\boldsymbol{p}}_{2,\ell,j} = \begin{cases} \mathbb{1}_{\ell=j} & \text{when } \ell \leq d \\ 0 & \text{when } \ell > d \end{cases}. \tag{2}$$

Under the above construction, we obtain that

$$\boldsymbol{Q}_1^\top \boldsymbol{K}_1 \boldsymbol{H} = \begin{bmatrix} I_d & \boldsymbol{0} \\ \boldsymbol{0} & \boldsymbol{0} \end{bmatrix} \in \mathbb{R}^{D\times N}, \quad \boldsymbol{Q}_2^\top \boldsymbol{K}_2 \boldsymbol{H} = \begin{bmatrix} -I_d & \boldsymbol{0} \\ \boldsymbol{0} & \boldsymbol{0} \end{bmatrix} \in \mathbb{R}^{D\times N},$$

$$\sigma(\boldsymbol{H}^\top \boldsymbol{Q}_1^\top \boldsymbol{K}_1 \boldsymbol{H}) + \sigma(\boldsymbol{H}^\top \boldsymbol{Q}_2^\top \boldsymbol{K}_2 \boldsymbol{H}) = \begin{bmatrix} \boldsymbol{X}^\top, \boldsymbol{0} \end{bmatrix} \in \mathbb{R}^{N\times N}.$$

We further obtain that

$$\frac{1}{N} \sum_{m=1}^M (\boldsymbol{V}_m \boldsymbol{H}) \times \sigma\left((\boldsymbol{Q}_m \boldsymbol{H})^\top (\boldsymbol{K}_m \boldsymbol{H})\right) = \begin{bmatrix} \boldsymbol{0} & \boldsymbol{0} \\ \boldsymbol{X}\boldsymbol{X}^\top \in \mathbb{R}^{d\times d} & \boldsymbol{0} \\ \boldsymbol{0} & \boldsymbol{0} \end{bmatrix} \in \mathbb{R}^{D\times D}.$$

Therefore, the output is given by $\tilde{\boldsymbol{H}}^{cov} = \begin{bmatrix} \boldsymbol{X} \\ \boldsymbol{X}\boldsymbol{X}^\top, \boldsymbol{0} \\ \tilde{\boldsymbol{p}}_{2,1}, \ldots, \tilde{\boldsymbol{p}}_{2,N} \\ \tilde{\boldsymbol{p}}_{\ell,1}, \ldots, \tilde{\boldsymbol{p}}_{\ell,N} \end{bmatrix}.$

#### 2. The Power Iteration.

Then we consider constructing a single attention layer that approximates the power iteration. This step involves two important operations: (1) Obtaining the vector given by $\boldsymbol{X}\boldsymbol{X}^\top\boldsymbol{v}$. (2) Approximation of the value of the inverse norm given by $1/\|\boldsymbol{X}\boldsymbol{X}^\top\boldsymbol{v}\|_2$. We show that one can use the multihead Relu Transformer to achieve both goals simulatenously, whose parameters are given by

$$\boldsymbol{V}_1^{pow,1} = -\boldsymbol{V}_2^{pow,1} = \begin{bmatrix} \boldsymbol{0}_{(3d+1)\times(2d+1)} & \boldsymbol{0} & \boldsymbol{0} \\ \boldsymbol{0}_{(d)\times(2d+1)} & I_d & \boldsymbol{0} \\ \boldsymbol{0} & \boldsymbol{0} & \boldsymbol{0} \end{bmatrix},$$

$$\boldsymbol{Q}_1^{pow,1} = -\boldsymbol{Q}_1^{pow,1} = \begin{bmatrix} \boldsymbol{0}_{(d+1)\times(d+1)} & \boldsymbol{0} & \boldsymbol{0} \\ \boldsymbol{0}_{d\times(d+1)} & I_d & \boldsymbol{0} \\ \boldsymbol{0} & \boldsymbol{0} & \boldsymbol{0} \end{bmatrix},$$

$$\boldsymbol{K}_1^{pow,1} = \boldsymbol{K}_2^{pow,1} = \begin{bmatrix} \boldsymbol{0}_{(3d+1)\times(3d+1)} & \boldsymbol{0} & \boldsymbol{0} \\ \boldsymbol{0}_{d\times(3d+1)} & I_d & \boldsymbol{0} \\ \boldsymbol{0} & \boldsymbol{0} & \boldsymbol{0} \end{bmatrix}, \qquad \tilde{\boldsymbol{p}}_{4,j} = \boldsymbol{0} \text{ for all } j \in [N].$$

Given the above formulation, we are able to show that

$$\boldsymbol{Q}_2^{pow,1}\tilde{\boldsymbol{H}}^{cov} = -\boldsymbol{Q}_1^{pow,1}\tilde{\boldsymbol{H}}^{cov} = \begin{bmatrix} \boldsymbol{0}_{(2d+1)\times N} \\ \boldsymbol{X}\boldsymbol{X}^\top, \boldsymbol{0} \\ \boldsymbol{0} \end{bmatrix},$$

$$\boldsymbol{K}_2^{pow,1}\tilde{\boldsymbol{H}}^{cov} = \boldsymbol{K}_1^{pow,1}\tilde{\boldsymbol{H}}^{cov} = \begin{bmatrix} \boldsymbol{0}_{2d+1} \\ \tilde{\boldsymbol{p}}_{3,1}, \boldsymbol{0} \\ \boldsymbol{0} \end{bmatrix},$$

which implies that

$$\sum_{m\in\{1,2\}} \sigma((\boldsymbol{Q}_m^{pow,1}\tilde{\boldsymbol{H}}^{cov})^\top \boldsymbol{K}_m^{pow,1}\tilde{\boldsymbol{H}}^{cov}) = \begin{bmatrix} \boldsymbol{0} & \boldsymbol{0}_{d\times(2d+1)} \\ \boldsymbol{X}\boldsymbol{X}^\top\tilde{\boldsymbol{p}}_{3,1} & \boldsymbol{0}_{d\times(N-1)} \\ \boldsymbol{0} & \boldsymbol{0} \end{bmatrix}.$$

Then we can show that

$$\tilde{\boldsymbol{H}}^{pow,1} - \tilde{\boldsymbol{H}}^{cov} = \sum_{m\in\{1,2\}} \boldsymbol{V}_m^{pow,1}\tilde{\boldsymbol{H}}^{cov} \times \sigma((\boldsymbol{Q}_m^{pow,1}\tilde{\boldsymbol{H}}^{cov})^\top \boldsymbol{K}_m^{pow,1}\tilde{\boldsymbol{H}}^{cov})$$

$$= \begin{bmatrix} \boldsymbol{0}_{3d+1} \\ \boldsymbol{X}\boldsymbol{X}^\top\tilde{\boldsymbol{p}}_{3,1}, \boldsymbol{0}_{d\times(N-1)} \\ \boldsymbol{0} \end{bmatrix}.$$

Therefore, we conclude that the output of the first power iteration layer is given by

$$\tilde{\boldsymbol{H}}^{pow,1} = \begin{bmatrix} \boldsymbol{X} \\ \tilde{\boldsymbol{y}}^\top \\ \boldsymbol{X}\boldsymbol{X}^\top, \boldsymbol{0} \\ \tilde{\boldsymbol{p}}_{2,1}, \ldots, \tilde{\boldsymbol{p}}_{2,N} \\ \tilde{\boldsymbol{p}}_{3,1}, \ldots, \tilde{\boldsymbol{p}}_{3,N} \\ \boldsymbol{X}\boldsymbol{X}^\top\tilde{\boldsymbol{p}}_{3,1}, \boldsymbol{0} \\ \tilde{\boldsymbol{p}}_{5,1}, \ldots, \tilde{\boldsymbol{p}}_{5,N} \\ \vdots \\ \tilde{\boldsymbol{p}}_{\ell,1}, \ldots, \tilde{\boldsymbol{p}}_{\ell,N} \end{bmatrix}.$$

Then, using lemma B.2, we design an extra attention layer that performs the normalizing procedure, with the following parameters for all $m \in [M]$,

$$\boldsymbol{V}_m^{pow,2} = \begin{bmatrix} \boldsymbol{0}_{d\times(4d+1)} & c_m\boldsymbol{I}_d & \boldsymbol{0} \\ \boldsymbol{0} & \boldsymbol{0} & \boldsymbol{0} \end{bmatrix}, \qquad \boldsymbol{Q}_m^{pow,2} = \begin{bmatrix} \boldsymbol{0}_{d\times(2d+1)} & \boldsymbol{I}_d & \boldsymbol{0} \\ \boldsymbol{0} & \boldsymbol{0} & \boldsymbol{0} \end{bmatrix},$$

$$\boldsymbol{K}_m^{pow,2} = \begin{bmatrix} \boldsymbol{0}_{1\times(3d+1)} & \boldsymbol{a}_m^\top & \boldsymbol{0} \\ \vdots & & \\ \boldsymbol{0}_{1\times(3d+1)} & \boldsymbol{a}_m^\top & \boldsymbol{0} \\ \boldsymbol{0}_{(D-d)\times(3d+1)} & \boldsymbol{0} & \boldsymbol{0} \end{bmatrix}.$$

Under the above construction, we obtain that

$$(\boldsymbol{Q}_m^{pow,2}\tilde{\boldsymbol{H}}^{pow,1})^\top = \begin{bmatrix} \boldsymbol{I}_{d\times d} & \boldsymbol{0} \\ \boldsymbol{0} & \boldsymbol{0} \end{bmatrix}, \qquad \boldsymbol{K}_m^{pow,2}\tilde{\boldsymbol{H}}^{pow,1} = \begin{bmatrix} \boldsymbol{a}_m^\top \boldsymbol{X}\boldsymbol{X}^\top \tilde{\boldsymbol{p}}_{3,1} & \boldsymbol{0} \\ \vdots \\ \boldsymbol{a}_m^\top \boldsymbol{X}\boldsymbol{X}^\top \tilde{\boldsymbol{p}}_{3,1} & \boldsymbol{0} \\ \boldsymbol{0}_{(D-d)\times 1} & \boldsymbol{0} \end{bmatrix}.$$

Then, given $\boldsymbol{V}_m^{pow,2}$ we can show that under the condition given by lemma B.2, we have

$$\left\| \sum_{m=1}^M \boldsymbol{V}_m^{pow,2}\tilde{\boldsymbol{H}}^{pow,1}\sigma\left((\boldsymbol{Q}_m^{pow,2}\tilde{\boldsymbol{H}}^{pow,1})^\top(\boldsymbol{K}_m^{pow,2}\tilde{\boldsymbol{H}}^{pow,1})\right) - \begin{bmatrix} \boldsymbol{0}_{4d+1} \\ \frac{\boldsymbol{X}\boldsymbol{X}^\top \tilde{\boldsymbol{p}}_{3,1}}{\|\boldsymbol{X}\boldsymbol{X}^\top \tilde{\boldsymbol{p}}_{3,1}\|_2} - \boldsymbol{X}\boldsymbol{X}^\top \tilde{\boldsymbol{p}}_{3,1}, \boldsymbol{0} \\ \boldsymbol{0} \end{bmatrix} \right\|_\infty$$

$$< \epsilon,$$

Moreover, we can further achieve that

$$\left\| \sum_{m=1}^M \boldsymbol{V}_m^{pow,2j}\tilde{\boldsymbol{H}}^{pow,1}\sigma\left((\boldsymbol{Q}_m^{pow,2}\tilde{\boldsymbol{H}}^{pow,1})^\top(\boldsymbol{K}_m^{pow,2}\tilde{\boldsymbol{H}}^{pow,1})\right) - \begin{bmatrix} \boldsymbol{0}_{4d+1} \\ \frac{\boldsymbol{X}\boldsymbol{X}^\top \tilde{\boldsymbol{p}}_{3,1}}{\|\boldsymbol{X}\boldsymbol{X}^\top \tilde{\boldsymbol{p}}_{3,1}\|_2} - \boldsymbol{X}\boldsymbol{X}^\top \tilde{\boldsymbol{p}}_{3,1}, \boldsymbol{0} \\ \boldsymbol{0} \end{bmatrix} \right\|_2$$
$$< \epsilon\|\boldsymbol{X}\boldsymbol{X}^\top \tilde{\boldsymbol{p}}_{3,1}\|_2.$$

Hence, using the fact that $\tilde{\boldsymbol{H}}^{pow,2} = \tilde{\boldsymbol{H}}^{pow,1} + \sum_{i=1}^m \boldsymbol{V}_m^{pow,2}\tilde{\boldsymbol{H}}^{pow,1}\sigma\left((\boldsymbol{Q}_m^{pow,2}\tilde{\boldsymbol{H}}^{pow,1})^\top(\boldsymbol{K}_m^{pow,2}\tilde{\boldsymbol{H}}^{pow,1})\right)$,
we obtain that

$$\left\| \tilde{\boldsymbol{H}}^{pow,2} - \begin{bmatrix} \boldsymbol{X} \\ \tilde{\boldsymbol{y}} \\ \boldsymbol{X}\boldsymbol{X}^\top, \boldsymbol{0} \\ \tilde{\boldsymbol{p}}_{2,1}, \ldots, \tilde{\boldsymbol{p}}_{2,N} \\ \tilde{\boldsymbol{p}}_{3,1}, \ldots, \tilde{\boldsymbol{p}}_{3,N} \\ \frac{\boldsymbol{X}\boldsymbol{X}^\top \tilde{\boldsymbol{p}}_{3,1}}{\|\boldsymbol{X}\boldsymbol{X}^\top \tilde{\boldsymbol{p}}_{3,1}\|_2}, \ldots \boldsymbol{0} \\ \vdots \end{bmatrix} \right\|_2 < \epsilon\|\boldsymbol{X}\boldsymbol{X}^\top \tilde{\boldsymbol{p}}_{3,1}\|_2.$$

Then we construct another attention layer, which performs similar calculations as that of $pow,1$ but switch the rows of $\tilde{\boldsymbol{p}}_{3,1}$ with that of $\frac{\boldsymbol{X}\boldsymbol{X}^\top \tilde{\boldsymbol{p}}_{3,1}}{\|\boldsymbol{X}\boldsymbol{X}^\top \tilde{\boldsymbol{p}}_{3,1}\|_2}$. Our construction for the third layer is given by

$$\boldsymbol{V}_1^{pow,3} = -\boldsymbol{V}_2^{pow,3} = \begin{bmatrix} \boldsymbol{0}_{(3d+1)\times(2d+1)} & \boldsymbol{0} & \boldsymbol{0} \\ \boldsymbol{0}_{d\times(2d+1)} & \boldsymbol{I}_d & \boldsymbol{0} \\ \boldsymbol{0} & \boldsymbol{0} & \boldsymbol{0} \end{bmatrix},$$

$$\boldsymbol{Q}_1^{pow,3} = -\boldsymbol{Q}_2^{pow,3} = \begin{bmatrix} \boldsymbol{0}_{(3d+1)\times(d+1)} & \boldsymbol{0} & \boldsymbol{0} \\ \boldsymbol{0}_{d\times(d+1)} & \boldsymbol{I}_d & \boldsymbol{0} \\ \boldsymbol{0} & \boldsymbol{0} & \boldsymbol{0} \end{bmatrix},$$

$$\boldsymbol{K}_1^{pow,3} = \boldsymbol{K}_2^{pow,3} = \begin{bmatrix} \boldsymbol{0}_{(4d+1)\times(4d+1)} & \boldsymbol{0} & \boldsymbol{0} \\ \boldsymbol{0}_{d\times(4d+1)} & \boldsymbol{I}_d & \boldsymbol{0} \\ \boldsymbol{0} & \boldsymbol{0} & \boldsymbol{0} \end{bmatrix}, \qquad \tilde{\boldsymbol{p}}_{4,j} = \boldsymbol{0} \text{ for all } j\in[N].$$

Given the above construction, we can show that

$$\boldsymbol{Q}_2^{pow,3}\tilde{\boldsymbol{H}}^{pow,2} = -\boldsymbol{Q}_1^{pow,3}\tilde{\boldsymbol{H}}^{pow,2} = \begin{bmatrix} \boldsymbol{0}_{(3d+1)\times N} \\ \boldsymbol{0} & \boldsymbol{X}\boldsymbol{X}^\top & \boldsymbol{0} \\ \boldsymbol{0} \end{bmatrix}, \quad \boldsymbol{K}_2^{pow,3}\tilde{\boldsymbol{H}}^{pow,2} = \boldsymbol{K}_1^{pow,3}\tilde{\boldsymbol{H}}^{pow,2},$$

$$\left\| \boldsymbol{K}_2^{pow,3}\tilde{\boldsymbol{H}}^{pow,2} - \begin{bmatrix} \boldsymbol{0}_{(3d+1)\times N} \\ \frac{\boldsymbol{X}\boldsymbol{X}^\top \tilde{\boldsymbol{p}}_{3,1}}{\|\boldsymbol{X}\boldsymbol{X}^\top \tilde{\boldsymbol{p}}_{3,1}\|_2}, \boldsymbol{0} \\ \boldsymbol{0} \end{bmatrix} \right\|_2 \le \epsilon\|\boldsymbol{X}\boldsymbol{X}^\top \tilde{\boldsymbol{p}}_{3,1}\|_2.$$

Then, using the fact that given $\boldsymbol{x}_1, \boldsymbol{x}_2$ with $\|\boldsymbol{x}_1 - \boldsymbol{x}_2\|_2 \leq \delta_0$, we have $\|\boldsymbol{X}\boldsymbol{X}^\top(\boldsymbol{x}_1 - \boldsymbol{x}_2)\|_2 \leq \|\boldsymbol{X}\boldsymbol{X}^\top\|_2 \delta_0$. Hence, collecting the above pieces, we have

$$\left\| \sum_{m=1}^{2} \boldsymbol{V}_m^{pow,3} \tilde{\boldsymbol{H}}^{pow,2} \sigma\left( (\boldsymbol{Q}_2^{pow,3} \tilde{\boldsymbol{H}}^{pow,2})^\top \boldsymbol{K}_2^{pow,3} \tilde{\boldsymbol{H}}^{pow,2} \right) - \begin{bmatrix} \boldsymbol{0}_{(3d+1)\times N} \\ \frac{(\boldsymbol{X}\boldsymbol{X}^\top)^2 \tilde{\boldsymbol{p}}_{3,1}}{\|\boldsymbol{X}\boldsymbol{X}^\top \tilde{\boldsymbol{p}}_{3,1}\|_2} - \frac{\boldsymbol{X}\boldsymbol{X}^\top \tilde{\boldsymbol{p}}_{3,1}}{\|\boldsymbol{X}\boldsymbol{X}^\top \tilde{\boldsymbol{p}}_{3,1}\|_2}, \boldsymbol{0} \\ \boldsymbol{0} \end{bmatrix} \right\|_2$$

$$\leq \epsilon \|\boldsymbol{X}\boldsymbol{X}^\top\|_2 \left\| \boldsymbol{X}\boldsymbol{X}^\top \tilde{\boldsymbol{p}}_{3,1} \right\|_2.$$

Henceforth, one can further show that $\left\| \tilde{\boldsymbol{H}}^{pow,3} - \begin{bmatrix} \boldsymbol{X} \\ \tilde{\boldsymbol{y}} \\ \boldsymbol{X}\boldsymbol{X}^\top, \boldsymbol{0} \\ \tilde{\boldsymbol{p}}_{2,1}, \ldots, \tilde{\boldsymbol{p}}_{2,N} \\ \tilde{\boldsymbol{p}}_{3,1}, \ldots, \tilde{\boldsymbol{p}}_{3,N} \\ \frac{(\boldsymbol{X}\boldsymbol{X}^\top)^2 \tilde{\boldsymbol{p}}_{3,1}}{\|\boldsymbol{X}\boldsymbol{X}^\top \tilde{\boldsymbol{p}}_{3,1}\|_2}, \boldsymbol{0} \\ \tilde{\boldsymbol{p}}_{5,1}, \ldots, \tilde{\boldsymbol{p}}_{5,N} \\ \vdots \\ \tilde{\boldsymbol{p}}_{\ell,1}, \ldots, \tilde{\boldsymbol{p}}_{\ell,N} \end{bmatrix} \right\|_2 \leq$

$\epsilon \|\boldsymbol{X}\boldsymbol{X}^\top\|_2 \|\boldsymbol{X}\boldsymbol{X}^\top \tilde{\boldsymbol{p}}_{3,1}\|_2$. Consider we are doing in total of $\tau$ power iterations, we can set for all $\tau \in \mathbb{N}^*$,

$$\boldsymbol{V}_m^{pow,2\tau+1} = \boldsymbol{V}_m^{pow,3}, \quad \boldsymbol{Q}_m^{pow,2\tau+1} = \boldsymbol{Q}_m^{pow,3}, \quad \boldsymbol{K}_m^{pow,2\tau+1} = \boldsymbol{K}_m^{pow,3},$$
$$\boldsymbol{V}_m^{pow,2\tau+2} = \boldsymbol{V}_m^{pow,4}, \quad \boldsymbol{Q}_m^{pow,2\tau+2} = \boldsymbol{Q}_m^{pow,4}, \quad \boldsymbol{K}_m^{pow,2\tau+2} = \boldsymbol{K}_m^{pow,4}.$$

Therefore, taking another layer of normalization, we can show that

$$\left\| \tilde{\boldsymbol{H}}^{pow,3} - \begin{bmatrix} \boldsymbol{X} \\ \tilde{\boldsymbol{y}} \\ \boldsymbol{X}\boldsymbol{X}^\top, \boldsymbol{0} \\ \tilde{\boldsymbol{p}}_{2,1}, \ldots, \tilde{\boldsymbol{p}}_{2,N} \\ \tilde{\boldsymbol{p}}_{3,1}, \ldots, \tilde{\boldsymbol{p}}_{3,N} \\ \frac{(\boldsymbol{X}\boldsymbol{X}^\top)^2 \tilde{\boldsymbol{p}}_{3,1}}{\|\boldsymbol{X}\boldsymbol{X}^\top \tilde{\boldsymbol{p}}_{3,1}\|_2^2}, \boldsymbol{0} \\ \tilde{\boldsymbol{p}}_{5,1}, \ldots, \tilde{\boldsymbol{p}}_{5,N} \\ \vdots \\ \tilde{\boldsymbol{p}}_{\ell,1}, \ldots, \tilde{\boldsymbol{p}}_{\ell,N} \end{bmatrix} \right\|_2 \leq 2\epsilon \|\boldsymbol{X}\boldsymbol{X}^\top\|_2.$$

Then, using the sublinearity of errors, we can show that for $\tau \in \mathbb{N}$,

$$\left\| \tilde{\boldsymbol{H}}^{pow,2\tau+2} - \begin{bmatrix} \boldsymbol{X} \\ \tilde{\boldsymbol{y}} \\ \boldsymbol{X}\boldsymbol{X}^\top, \boldsymbol{0} \\ \tilde{\boldsymbol{p}}_{2,1}, \ldots, \tilde{\boldsymbol{p}}_{2,N} \\ \tilde{\boldsymbol{p}}_{3,1}, \ldots, \tilde{\boldsymbol{p}}_{3,N} \\ \tilde{\boldsymbol{p}}_{3,1}^{(\tau)}, \boldsymbol{0} \\ \tilde{\boldsymbol{p}}_{5,1}, \ldots, \tilde{\boldsymbol{p}}_{5,N} \\ \vdots \\ \tilde{\boldsymbol{p}}_{\ell,1}, \ldots, \tilde{\boldsymbol{p}}_{\ell,N} \end{bmatrix} \right\|_\infty \leq \tau\epsilon \|\boldsymbol{X}\boldsymbol{X}^\top\|_2, \quad \tilde{\boldsymbol{p}}_{3,1}^{(\tau)} = \frac{\boldsymbol{X}\boldsymbol{X}^\top \tilde{\boldsymbol{p}}_{3,1}^{(\tau-1)}}{\left\| \boldsymbol{X}\boldsymbol{X}^\top \tilde{\boldsymbol{p}}_{3,1}^{(\tau-1)} \right\|_2}, \quad \tilde{\boldsymbol{p}}_{3,1}^{(0)} = \tilde{\boldsymbol{p}}_{3,1}.$$

If we denote $\boldsymbol{v}_i$ as the eigenvector corresponds to the $i$ th largest eigenvalue of $\boldsymbol{X}\boldsymbol{X}^\top$. Let the eigenvalues of $\boldsymbol{X}\boldsymbol{X}^\top$ be denoted by $\lambda_1 > \lambda_2 > \cdots > \lambda_n$. Given $|\tilde{\boldsymbol{p}}_{3,1}^\top \boldsymbol{v}_1| > \delta$ and $|\sqrt{\lambda_1} - \sqrt{\lambda_2}| = \Omega(1)$. Theorem 3.11 in (Blum et al., 2020) page 53 shows that given $k = \frac{\log(1/\epsilon_0 \delta)}{2\epsilon_0}$ and $\|\tilde{\boldsymbol{p}}_{3,1}^{(\tau)}\|_2 = \|\boldsymbol{v}_1\|_2 = 1$, one immediately obtains that

$$\tilde{\boldsymbol{p}}_{3,1}^{(\tau),\top} \boldsymbol{v}_1 \geq 1 - \epsilon_0, \qquad \|\tilde{\boldsymbol{p}}_{3,1}^{(\tau)} - \boldsymbol{v}_1\|_2 = \sqrt{2 - 2\boldsymbol{v}_1^\top \tilde{\boldsymbol{p}}_{3,1}^{(\tau)}} = \sqrt{2\epsilon_0}.$$

And we also consider the approximation of the maximum eigenvalue. Note that using $\|\boldsymbol{v}_1\|_2 = 1$, we have

$$\|\boldsymbol{X}\boldsymbol{X}^\top\|_2 = \|\boldsymbol{X}\boldsymbol{X}^\top \boldsymbol{v}_1\|_2 = \left\|\boldsymbol{X}\boldsymbol{X}^\top \tilde{\boldsymbol{p}}_{3,1}^{(\tau)} + \boldsymbol{X}\boldsymbol{X}^\top(\boldsymbol{v}_1 - \tilde{\boldsymbol{p}}_{3,1}^{(\tau)})\right\|_2$$

$$\leq \|\boldsymbol{X}\boldsymbol{X}^\top \tilde{\boldsymbol{p}}_{3,1}^{(\tau)}\|_2 + \|\boldsymbol{X}\boldsymbol{X}^\top(\boldsymbol{v}_1 - \tilde{\boldsymbol{p}}_{3,1}^{(\tau)})\|_2$$

$$\leq \|\boldsymbol{X}\boldsymbol{X}^\top \tilde{\boldsymbol{p}}_{3,1}^{(\tau)}\|_2 + \|\boldsymbol{X}\boldsymbol{X}^\top\|_2 \|\boldsymbol{v}_1 - \tilde{\boldsymbol{p}}_{3,1}^{(\tau)}\|_2.$$

Similarly we can also derive that $\|\boldsymbol{X}\boldsymbol{X}^\top\|_2 \geq \|\boldsymbol{X}\boldsymbol{X}^\top \tilde{\boldsymbol{p}}_{3,1}^{(\tau)}\|_2 - \|\boldsymbol{X}\boldsymbol{X}^\top\|_2 \|\boldsymbol{v}_1 - \tilde{\boldsymbol{p}}_{3,1}\|_2$. Then we show that

$$\left|\|\boldsymbol{X}\boldsymbol{X}^\top\|_2 - \|\boldsymbol{X}\boldsymbol{X}^\top \tilde{\boldsymbol{p}}_{3,1}^{(\tau)}\|_2\right| \leq \|\boldsymbol{X}\boldsymbol{X}^\top\|_2 \|\boldsymbol{v}_1 - \tilde{\boldsymbol{p}}_{3,1}^{(\tau)}\|_2 \leq \sqrt{2\epsilon_0}\|\boldsymbol{X}\boldsymbol{X}^\top\|_2.$$

### 3. The Removal of Principle Eigenvectors.

After $\tau$ iterates on the power method, we need to remove the principle term from the matrix $\boldsymbol{X}\boldsymbol{X}^\top$, achieved through two important steps: (1) The computation of the estimated eigenvalue $\|\boldsymbol{X}\boldsymbol{X}^\top \tilde{\boldsymbol{p}}_{3,1}\|_2$. (2) The construction of the low rank update $\tilde{\boldsymbol{p}}_{3,1}\tilde{\boldsymbol{p}}_{3,1}^\top$. For step (1), we consider the following construction:

$$\boldsymbol{V}_1^{rpe,1} = -\boldsymbol{V}_2^{rpe,1} = \begin{bmatrix} \boldsymbol{0}_{(3d+1)\times(2d+1)} & \boldsymbol{0} & \boldsymbol{0} \\ \boldsymbol{0}_{d\times(2d+1)} & I_d & \boldsymbol{0} \\ \boldsymbol{0} & \boldsymbol{0} & \boldsymbol{0} \end{bmatrix}, \quad \boldsymbol{Q}_1^{rpe,1} = -\boldsymbol{Q}_2^{rpe,1} = \begin{bmatrix} \boldsymbol{0}_{(d+1)\times(d+1)} & \boldsymbol{0} & \boldsymbol{0} \\ \boldsymbol{0}_{d\times(d+1)} & I_d & \boldsymbol{0} \\ \boldsymbol{0} & \boldsymbol{0} & \boldsymbol{0} \end{bmatrix},$$

$$\boldsymbol{K}_1^{rpe,1} = \boldsymbol{K}_2^{rpe,1} = \begin{bmatrix} \boldsymbol{0}_{(4d+1)\times(4d+1)} & \boldsymbol{0} & \boldsymbol{0} \\ \boldsymbol{0}_{d\times(4d+1)} & I_d & \boldsymbol{0} \\ \boldsymbol{0} & \boldsymbol{0} & \boldsymbol{0} \end{bmatrix}.$$

Note that the above construction is similar to the first layer of the power method. Under this construction, we can show that

$$\tilde{\boldsymbol{H}}^{rpe,1} = \tilde{\boldsymbol{H}}^{pow,2\tau+2} + \sum_{m\in\{1,2\}} \boldsymbol{V}_m^{rpe,1}\sigma((\boldsymbol{Q}_m^{rpe,1}\tilde{\boldsymbol{H}}^{pow,2\tau+2})^\top(\boldsymbol{K}_m^{rpe,1}\tilde{\boldsymbol{H}}^{pow,2\tau+2})),$$

$$\left\|\tilde{\boldsymbol{H}}^{rpe,1} - \underbrace{\begin{bmatrix} \boldsymbol{X} \\ \tilde{\boldsymbol{y}} \\ \boldsymbol{X}\boldsymbol{X}^\top, \boldsymbol{0} \\ \tilde{\boldsymbol{p}}_{2,1},\ldots,\tilde{\boldsymbol{p}}_{2,N} \\ \tilde{\boldsymbol{p}}_{3,1},\ldots,\tilde{\boldsymbol{p}}_{3,N} \\ \tilde{\boldsymbol{p}}_{3,1}^{(\tau)}, \boldsymbol{0} \\ \boldsymbol{X}\boldsymbol{X}^\top \tilde{\boldsymbol{p}}_{3,N}^{(\tau)}, \boldsymbol{0} \\ \tilde{\boldsymbol{p}}_{6,1},\ldots,\tilde{\boldsymbol{p}}_{6,N} \\ \vdots \\ \tilde{\boldsymbol{p}}_{\ell,1},\ldots,\tilde{\boldsymbol{p}}_{\ell,N} \end{bmatrix}}_{=:\boldsymbol{H}^{rpe,1}}\right\|_2 \leq C\tau\epsilon\|\boldsymbol{X}\boldsymbol{X}^\top\|_2^2, \qquad \tilde{\boldsymbol{p}}_{5,i} = \boldsymbol{0}, \quad \forall i \in [N]. \qquad (3)$$

Then, we construct the next layer, using the notations in lemma B.2, for $M \geq \|\boldsymbol{X}\boldsymbol{X}^\top\|_2^d \frac{C(d)}{\epsilon^2}$ for all $m \in [M]$ we have

$$\boldsymbol{V}_m^{rpe,2} = \begin{bmatrix} \boldsymbol{0}_{d\times(4d+1)} & d_m I_d & \boldsymbol{0} \\ \boldsymbol{0} & \boldsymbol{0} & \boldsymbol{0} \end{bmatrix}, \qquad \boldsymbol{Q}_m^{rpe,2} = \begin{bmatrix} \boldsymbol{0}_{d\times(2d+1)} & I_d & \boldsymbol{0} \\ \boldsymbol{0} & \boldsymbol{0} & \boldsymbol{0} \end{bmatrix},$$

$$\boldsymbol{K}_m^{rpe,2} = \begin{bmatrix} \boldsymbol{0}_{1\times(5d+1)} & \boldsymbol{b}_m^\top & \boldsymbol{0} \\ \vdots & & \\ \boldsymbol{0}_{1\times(5d+1)} & \boldsymbol{b}_m^\top & \boldsymbol{0} \\ \boldsymbol{0}_{(D-d)\times(5d+1)} & \boldsymbol{0} & \boldsymbol{0} \end{bmatrix}.$$

Given the above construction, we subsequently show that

$$(\boldsymbol{Q}_m^{rpe,2}\tilde{\boldsymbol{H}}^{rpe,1})^\top = \begin{bmatrix} I_{d\times d} & \boldsymbol{0} \\ \boldsymbol{0} & \boldsymbol{0} \end{bmatrix}, \qquad \boldsymbol{K}_m^{rpe,2}\tilde{\boldsymbol{H}}^{rpe,1} = \begin{bmatrix} \boldsymbol{b}_m^\top \boldsymbol{X}\boldsymbol{X}^\top \tilde{\boldsymbol{p}}_{3,1}^{(\tau)} & \boldsymbol{0} \\ \vdots & \vdots \\ \boldsymbol{b}_m^\top \boldsymbol{X}\boldsymbol{X}^\top \tilde{\boldsymbol{p}}_{3,1}^{(\tau)} & \boldsymbol{0} \\ \boldsymbol{0}_{(D-d)\times 1} & \boldsymbol{0} \end{bmatrix}.$$

Hence, given the construction of $\boldsymbol{V}_m^{rpe,2}$, we can show that $\tilde{\boldsymbol{H}}^{rpe,2}$ satisfies

$$\tilde{\boldsymbol{H}}^{rpe,2} = \tilde{\boldsymbol{H}}^{rpe,1} + \sum_{m\in[M]} \boldsymbol{V}_m^{rpe,2}\tilde{\boldsymbol{H}}^{rpe,1} \times \sigma\left((\boldsymbol{K}_m^{rpe,2}\tilde{\boldsymbol{H}}^{rpe,1})^\top(\boldsymbol{Q}_m^{rpe}\tilde{\boldsymbol{H}}^{rpe,1})\right)$$

$$= \underbrace{\boldsymbol{H}^{rpe,1} + \sum_{m\in[M]} \boldsymbol{V}_m^{rpe,2}\boldsymbol{H}^{rpe,1} \times \sigma\left((\boldsymbol{K}_m^{rpe,2}\boldsymbol{H}^{rpe,1})^\top(\boldsymbol{Q}_m^{rpe,2}\boldsymbol{H}^{rpe,1})\right)}_{=:\widehat{\boldsymbol{H}}^{rpe,1}}$$

$$+ \left(\tilde{\boldsymbol{H}}^{rpe,1} - \boldsymbol{H}^{rpe,1}\right) + \sum_{m\in[M]} \boldsymbol{V}_m^{rpe,2}\tilde{\boldsymbol{H}}^{rpe,1} \times \sigma\left((\boldsymbol{K}_m^{rpe,2}\tilde{\boldsymbol{H}}^{rpe,1})^\top \boldsymbol{Q}_m^{rpe,2}\tilde{\boldsymbol{H}}^{rpe,1}\right)$$

$$- \sum_{m\in[M]} \boldsymbol{V}_m^{rpe,2}\tilde{\boldsymbol{H}}^{rpe,1} \times \sigma\left((\boldsymbol{K}_m^{rpe,2}\tilde{\boldsymbol{H}}^{rpe,1})^\top \boldsymbol{Q}_m^{rpe,2}\tilde{\boldsymbol{H}}^{rpe,1}\right).$$

We note that by lemma B.2 we can show that

$$\left\|\widehat{\boldsymbol{H}}^{rpe,1} - \begin{bmatrix} \boldsymbol{X} \\ \tilde{\boldsymbol{y}} \\ \boldsymbol{X}\boldsymbol{X}^\top, \boldsymbol{0} \\ \tilde{\boldsymbol{p}}_{2,1}\ldots,\tilde{\boldsymbol{p}}_{2,N} \\ \tilde{\boldsymbol{p}}_{3,1}\ldots,\tilde{\boldsymbol{p}}_{3,N} \\ \tilde{\boldsymbol{p}}_{3,1}^{(\tau)}, \boldsymbol{0} \\ \|\boldsymbol{X}\boldsymbol{X}^\top\tilde{\boldsymbol{p}}_{3,N}^{(\tau)}\|_2^{\frac{1}{2}}\tilde{\boldsymbol{p}}_{3,1}^{(\tau)}, \boldsymbol{0} \\ \vdots \\ \tilde{\boldsymbol{p}}_{\ell,1},\ldots\tilde{\boldsymbol{p}}_{\ell,N} \end{bmatrix}\right\|_2 \le C\tau\epsilon\|\boldsymbol{X}\boldsymbol{X}^\top\|_2^2.$$

Then the rest of the proof focuses on showing that the rest of the terms are small. Note that using equation 3, we show that

$$\left\|\tilde{\boldsymbol{H}}^{rpe,1} - \boldsymbol{H}^{rpe,1}\right\|_2 \le \tau\epsilon\|\boldsymbol{X}\boldsymbol{X}^\top\|_2^2.$$

And for the last term, we can show that

$$\Big\|\sum_{m\in[M]} \boldsymbol{V}_m^{rpe,2}\tilde{\boldsymbol{H}}^{rpe,1} \times \sigma\left((\boldsymbol{K}_m^{rpe,2}\tilde{\boldsymbol{H}}^{rpe,1})^\top(\boldsymbol{Q}_m^{rpe,2}\tilde{\boldsymbol{H}}^{rpe,1})\right)$$

$$- \sum_{m\in[M]} \boldsymbol{V}_m^{rpe,2}\boldsymbol{H}^{rpe,1} \times \sigma\left((\boldsymbol{K}_m^{rpe,2}\boldsymbol{H}^{rpe,1})^\top(\boldsymbol{Q}_m^{rpe,2}\boldsymbol{H}^{rpe,1})\right)\Big\|_2$$

$$\le C\tau\epsilon\|\boldsymbol{X}\boldsymbol{X}^\top\|_2^2.$$

Collecting the above pieces, we finally show that

$$\left\|\tilde{\boldsymbol{H}}^{rpe,2} - \begin{bmatrix} \boldsymbol{X} \\ \boldsymbol{X}\boldsymbol{X}^\top, \boldsymbol{0} \\ \tilde{\boldsymbol{p}}_{2,1},\ldots,\tilde{\boldsymbol{p}}_{2,N} \\ \tilde{\boldsymbol{p}}_{3,1},\ldots,\tilde{\boldsymbol{p}}_{3,N} \\ \tilde{\boldsymbol{p}}_{3,1}^{(\tau)}, \boldsymbol{0} \\ \|\boldsymbol{X}\boldsymbol{X}^\top\tilde{\boldsymbol{p}}_{3,1}^{(\tau)}\|_2^{\frac{1}{2}}\tilde{\boldsymbol{p}}_{3,1}^{(\tau)}, \boldsymbol{0} \\ \tilde{\boldsymbol{p}}_{6,1},\ldots,\tilde{\boldsymbol{p}}_{6,N} \\ \vdots \\ \tilde{\boldsymbol{p}}_{\ell,1},\ldots,\tilde{\boldsymbol{p}}_{\ell,N} \end{bmatrix}\right\|_2 \le C\tau\epsilon\|\boldsymbol{X}\boldsymbol{X}^\top\|_2^2.$$

Then we construct another layer to remove the principle components from the matrix $\boldsymbol{X}\boldsymbol{X}^\top$, given by

$$-\boldsymbol{V}_1^{rpe,3} = \boldsymbol{V}_2^{rpe,3} = \begin{bmatrix} \boldsymbol{0}_{(d+1)\times(4d+1)} & \boldsymbol{0} & \boldsymbol{0} \\ \boldsymbol{0} & I_d & \boldsymbol{0} \\ \boldsymbol{0} & \boldsymbol{0} & \boldsymbol{0} \end{bmatrix}, \qquad \boldsymbol{Q}_1^{rpe,3} = -\boldsymbol{Q}_2^{rpe,3} = \begin{bmatrix} \boldsymbol{0}_{d\times(4d+1)} & I_d & \boldsymbol{0} \\ \boldsymbol{0} & \boldsymbol{0} & \boldsymbol{0} \end{bmatrix},$$

$$\boldsymbol{K}_1^{rpe,3} = \boldsymbol{K}_2^{rpe,3} = \begin{bmatrix} \boldsymbol{0}_{d\times(4d+1)} & I_d & \boldsymbol{0} \\ \boldsymbol{0} & \boldsymbol{0} & \boldsymbol{0} \end{bmatrix}.$$

Then we can show that

$$(\boldsymbol{Q}_1^{rpe,3}\tilde{\boldsymbol{H}}^{rpe,2})^\top = \begin{bmatrix} \|\boldsymbol{X}\boldsymbol{X}^\top\tilde{\boldsymbol{p}}_{3,1}^{(\tau)}\|_2^{\frac{1}{2}}\tilde{\boldsymbol{p}}_{3,1}^{(\tau),\top} & \boldsymbol{0} \\ \boldsymbol{0} & \boldsymbol{0} \end{bmatrix}, \qquad \boldsymbol{K}_1^{rpe,3}\tilde{\boldsymbol{H}}^{rpe,2} = \begin{bmatrix} \boldsymbol{0} & \boldsymbol{0} \\ I_d & \boldsymbol{0} \\ \boldsymbol{0} & \boldsymbol{0} \end{bmatrix}.$$

Then it is further noted that $-(\boldsymbol{Q}_2^{rpe,3}\tilde{\boldsymbol{H}}^{rpe,2})^\top\boldsymbol{K}_2^{rpe,3}\tilde{\boldsymbol{H}}^{rpe,2} = (\boldsymbol{Q}_1^{rpe,3}\tilde{\boldsymbol{H}}^{rpe,2})^\top\boldsymbol{K}_1^{rpe,3}\tilde{\boldsymbol{H}}^{rpe,2}$ satisfies

$$\left\| (\boldsymbol{Q}_1^{rpe,3}\tilde{\boldsymbol{H}}^{rpe,2})^\top\boldsymbol{K}_1^{rpe,3}\tilde{\boldsymbol{H}}^{rpe,2} - \begin{bmatrix} \|\boldsymbol{X}\boldsymbol{X}^\top\tilde{\boldsymbol{p}}_{3,1}^{(\tau)}\|_2^{\frac{1}{2}}\tilde{\boldsymbol{p}}_{3,1}^{(\tau),\top} & \boldsymbol{0} \\ \boldsymbol{0} & \boldsymbol{0} \end{bmatrix} \right\|_2 \leq C\tau\epsilon\|\boldsymbol{X}\boldsymbol{X}^\top\|_2^2.$$

And therefore, combining our construction for $\boldsymbol{V}_m$, it is noted that

$$\left\| \sum_{m=1}^2 \boldsymbol{V}_m\tilde{\boldsymbol{H}}^{rpe,2} \times \sigma((\boldsymbol{Q}_1^{rpe,3}\tilde{\boldsymbol{H}}^{rpe,2})^\top\boldsymbol{K}_1^{rpe,3}\tilde{\boldsymbol{H}}^{rpe,2}) - \begin{bmatrix} \boldsymbol{0}_{(d+1)\times N} \\ -\|\boldsymbol{X}\boldsymbol{X}^\top\tilde{\boldsymbol{p}}_{3,1}^{(\tau)}\|_2\tilde{\boldsymbol{p}}_{3,1}^{(\tau)}\tilde{\boldsymbol{p}}_{3,1}^{(\tau),\top}, \boldsymbol{0} \\ \boldsymbol{0} \end{bmatrix} \right\|_2$$
$$\leq C\tau\epsilon\|\boldsymbol{X}\boldsymbol{X}^\top\|_2^2.$$

Therefore, we can further show that

$$\tilde{\boldsymbol{H}}^{rpe,3} = \tilde{\boldsymbol{H}}^{rpe,2} + \sum_{m=1}^2 \boldsymbol{V}_m^{rpe,3}\tilde{\boldsymbol{H}}^{rpe,2} \times \sigma\left((\boldsymbol{Q}_m^{rpe,3}\tilde{\boldsymbol{H}}^{rpe,2})^\top\boldsymbol{K}_m^{rpe,3}\tilde{\boldsymbol{H}}^{rpe,2}\right)$$

satisfies

$$\left\| \tilde{\boldsymbol{H}}^{rpe,3} - \begin{bmatrix} \boldsymbol{X} \\ \boldsymbol{X}\boldsymbol{X}^\top - \|\boldsymbol{X}\boldsymbol{X}^\top\tilde{\boldsymbol{p}}_{3,1}^{(\tau)}\|_2\tilde{\boldsymbol{p}}_{3,1}^{(\tau)}\tilde{\boldsymbol{p}}_{3,1}^{(\tau),\top}, \boldsymbol{0} \\ \tilde{\boldsymbol{p}}_{2,1},\ldots,\tilde{\boldsymbol{p}}_{2,N} \\ \tilde{\boldsymbol{p}}_{3,1},\ldots,\tilde{\boldsymbol{p}}_{3,N} \\ \tilde{\boldsymbol{p}}_{3,1}^{(\tau)},\boldsymbol{0} \\ \tilde{\boldsymbol{p}}_{5,1},\ldots,\tilde{\boldsymbol{p}}_{5,N} \\ \vdots \\ \tilde{\boldsymbol{p}}_{\ell,1},\ldots,\tilde{\boldsymbol{p}}_{\ell,N} \end{bmatrix} \right\|_2 \leq C\tau\epsilon\|\boldsymbol{X}\boldsymbol{X}^\top\|_2^2.$$

And we can construct another layer to remove the term $\|\boldsymbol{X}\boldsymbol{X}^\top\tilde{\boldsymbol{p}}_{3,1}^{(\tau)}\|_2^{\frac{1}{2}}\tilde{\boldsymbol{p}}_{3,1}^{(\tau)}$, which is achieved by

$$-\boldsymbol{V}_1^{rpe,4} = \boldsymbol{V}_2^{rpe,4} = \begin{bmatrix} \boldsymbol{0}_{(4d+1)\times(4d+1)} & \boldsymbol{0} & \boldsymbol{0} \\ \boldsymbol{0}_{d\times(4d+1)} & I_d & \boldsymbol{0} \\ \boldsymbol{0} & \boldsymbol{0} & \boldsymbol{0} \end{bmatrix},$$

$$\boldsymbol{Q}_1^{rpe,4} = -\boldsymbol{Q}_2^{rpe,4} = \begin{bmatrix} \boldsymbol{0}_{(3d+1)\times D} \\ \boldsymbol{0}_{d\times(2d+1)} & I_d & \boldsymbol{0} \\ \boldsymbol{0} \end{bmatrix},$$

$$\boldsymbol{K}_1^{rpe,4} = \boldsymbol{K}_2^{rpe,4} = \begin{bmatrix} \boldsymbol{0}_{(3d+1)\times D} \\ \boldsymbol{0}_{d\times(2d+1)} & I_d & \boldsymbol{0} \\ \boldsymbol{0} \end{bmatrix}.$$

Using the above construction, we can further show that

$$\left\| \tilde{H}^{rpe,4} - \begin{bmatrix} X \\ XX^\top - \|XX^\top \tilde{p}_{3,1}^{(\tau)}\|_2 \tilde{p}_{3,1}^{(\tau)} \tilde{p}_{3,1}^{(\tau),\top}, 0 \\ \tilde{p}_{2,1}, \ldots, \tilde{p}_{2,N} \\ \tilde{p}_{3,1}, \ldots, \tilde{p}_{3,N} \\ \tilde{p}_{3,1}^{(\tau)}, 0 \\ \tilde{p}_{5,1}, \ldots, \tilde{p}_{5,N} \\ \vdots \\ \tilde{p}_{\ell,1}, \ldots, \tilde{p}_{\ell,N} \end{bmatrix} \right\|_2 \leq C\tau\epsilon \|XX^\top\|_2^2.$$

And then we proceed to recover the rest of the $k$ principle eigenvectors using similar model architecture given by the ones used by the Power Iterations. For the computation over the $\tau$-th eigenvector, we denote $\tilde{H}^{pow,\eta,1}$ till $\tilde{H}^{pow,\eta,\tau}$ to be the intermediate states corresponding to the $\eta$-th power iteration. We denote $\tilde{H}^{rpe,\eta,\tau_0}$ to be the output of $\eta$-th removal of principle eigenvector layers for the $\tau$-th eigenvector. Furthermore, we iteratively define

$$A_1 = XX^\top - \|XX^\top \tilde{p}_{3,1}^{(\tau)}\|_2 \tilde{p}_{3,1}^{(\tau)} \tilde{p}_{3,1}^{(\tau),\top}, \qquad A_{i+1} = A_i - \|A_i \tilde{p}_{3,i}^{(\tau)}\|_2 \tilde{p}_{3,i}^{(\tau)} \tilde{p}_{3,i}^{(\tau),\top}, \qquad \forall i \in [k].$$

Then, applying the subadditivity of the 2-norm, we can show that

$$\left\| \tilde{H}^{rpe,4,k} - \begin{bmatrix} X \\ A_{k+1}, 0 \\ \tilde{p}_{2,1}, \ldots, \tilde{p}_{2,N} \\ \tilde{p}_{3,1}, \ldots, \tilde{p}_{3,N} \\ \tilde{p}_{3,1}^{(\tau)}, 0 \\ \tilde{p}_{3,2}^{(\tau)}, 0 \\ \vdots \\ \tilde{p}_{3,k}^{(\tau)}, 0 \end{bmatrix} \right\|_2 \leq C\tau k\epsilon \|XX^\top\|_2^2.$$

For simplicity, we denote $\tilde{A} = \begin{bmatrix} X \\ A_{k+1}, 0 \\ \tilde{p}_{2,1}, \ldots, \tilde{p}_{2,N} \\ \tilde{p}_{3,1}, \ldots, \tilde{p}_{3,N} \end{bmatrix}$ and $\tilde{P} = \begin{bmatrix} \tilde{p}_{3,1}^{(\tau)} \\ \tilde{p}_{3,2}^{(\tau)} \\ \vdots \\ \tilde{p}_{3,k}^{(\tau)} \end{bmatrix}$ from here.

### 4. Finishing Up.

The finishing up phase considers constructing $\tilde{W}_0$ and $\tilde{W}_1$ that adjust the final output format. Our construction gives the following

$$\tilde{W}_0 = [0, I_{kd}], \qquad \tilde{W}_1 = \begin{bmatrix} 1 \\ 0_{N-1} \end{bmatrix}.$$

And we can show that

$$\left\| \tilde{W}_0 \tilde{H}^{rpe,4,k} \tilde{W}_1 - \begin{bmatrix} \tilde{p}_{3,1}^{(\tau)} \\ \tilde{p}_{3,2}^{(\tau)} \\ \vdots \\ \tilde{p}_{3,k}^{(\tau)} \end{bmatrix} \right\|_2 \leq C\tau k\epsilon \|XX^\top\|_2^2.$$

We further use the result given by lemma B.1, denote $a_\eta := \left\| v_\eta - \tilde{p}_{3,\eta}^{(\tau)} \right\|_2$, $\widehat{\lambda}_\eta = \left\| A_\eta \tilde{p}_{3,\eta}^{(\tau)} \right\|_2$, and $b_\eta := |\lambda_\eta - \widehat{\lambda}_\eta|$ for $\eta \in [k]$, we obtain that for all $\eta \geq 1$, given the number of iterations $\tau \geq C \frac{\log(1/\epsilon_0 \delta)}{2\epsilon_0}$ where the constant value $C$ depends on $d$,

$$a_{\eta+1} \leq \frac{\max_{i \in [\eta]} b_i + \sum_{i=1}^\eta 2\lambda_i a_i}{\Delta}, \qquad b_{\eta+1} \leq \frac{2\lambda_{\eta+1}}{\Delta} \left( \max_{i \in [\eta]} b_\eta + \sum_{i=1}^\eta 2\lambda_i a_i \right) + \lambda_{\eta+1} \sqrt{2\epsilon_0}.$$

Further note that the starting point is given by $a_1 \leq \sqrt{2\epsilon_0}$, $b_1 \leq \lambda_1\sqrt{2\epsilon_0}$. Introducing $A_\eta = \sum_{i=1}^{\eta} 2\lambda_i a_i$, we obtain that $A_{\eta+1} = \sum_{i=1}^{\eta+1} 2\lambda_i a_i = A_\eta + 2\lambda_{\eta+1}a_{\eta+1}$ which alternatively implies that

$$\frac{1}{2\lambda_{\eta+1}}(A_{\eta+1} - A_\eta) \leq \frac{\max_{i\in[\eta]} b_i + A_\eta}{\Delta}, \qquad b_{\eta+1} \leq \frac{2\lambda_{\eta+1}}{\Delta}\left(\max_{i\in[\eta]} b_\eta + A_\eta\right) + \lambda_{\eta+1}\sqrt{2\epsilon_0}.$$

We use the fact $\frac{\lambda_\eta}{\Delta} > 1$ for all $\eta \in [k]$ to show the following

$$A_{\eta+1} + \max_{i\in[\eta+1]} b_i \leq \frac{5\lambda_{\eta+1}}{\Delta}\left(A_\eta + \max_{i\in[\eta]} b_i\right) + \lambda_1\sqrt{2\epsilon_0}, \qquad A_1 + b_1 = 2\lambda_1\sqrt{2\epsilon_0},$$

which implies that

$$A_{\eta+1} + \max_{i\in[\eta+1]} b_i + \frac{\lambda_1\sqrt{2\epsilon_0}}{\frac{5\lambda_1}{\Delta} - 1} \leq \frac{5\lambda_1}{\Delta}\left(A_\eta + \max_{i\in[\eta]} b_i + \frac{\lambda_1\sqrt{2\epsilon_0}}{\frac{5\lambda_1}{\Delta} - 1}\right),$$

$$A_{\eta+1} + \max_{i\in[\eta+1]} b_i + \frac{\lambda_1\sqrt{2\epsilon_0}}{\frac{5\lambda_1}{\Delta} - 1} \leq \left(A_1 + b_1 + \frac{\lambda_1\sqrt{2\epsilon_0}}{\frac{5\lambda_1}{\Delta} - 1}\right)\prod_{i=1}^{\eta}\left(\frac{5\lambda_{i+1}}{\Delta}\right)$$

$$= \lambda_1\sqrt{2\epsilon_0}\left(2 + \frac{1}{\frac{5\lambda_1}{\Delta} - 1}\right)\prod_{i=1}^{\eta}\left(\frac{5\lambda_{i+1}}{\Delta}\right). \tag{4}$$

Therefore, applying the inequality given by equation 4 we can show that, for $\eta \leq k$, we have for all $\eta \in [k-1]$,

$$a_{\eta+1} \leq \frac{1}{\Delta}\left(\lambda_1\sqrt{2\epsilon_0}\left(2 + \frac{1}{\frac{5\lambda_1}{\Delta} - 1}\right)\prod_{i=1}^{\eta}\left(\frac{5\lambda_{i+1}}{\Delta}\right) - \frac{\lambda_1\sqrt{2\epsilon_0}}{\frac{5\lambda_1}{\Delta} - 1}\right),$$

$$b_{\eta+1} \leq \frac{2\lambda_\eta\lambda_1\sqrt{2\epsilon_0}}{\Delta}\left(2 + \frac{1}{\frac{5\lambda_1}{\Delta} - 1}\right)\prod_{i=1}^{\eta}\left(\frac{5\lambda_{i+1}}{\Delta}\right) + \lambda_{\eta+1}\sqrt{2\epsilon_0}.$$

Therefore collecting pieces, we conclude that there exists a transformer with number of layers $2\tau + 4k + 1$ and number of heads $M \leq \lambda_1^d \frac{C(d)}{\epsilon^2}$ such that the final output $\widehat{\boldsymbol{v}}_1, \ldots, \widehat{\boldsymbol{v}}_k$ given by the Transformer model satisfy $\forall \eta \in [k-1]$,

$$\|\widehat{\boldsymbol{v}}_{\eta+1} - \boldsymbol{v}_{\eta+1}\|_2 \leq C\tau\epsilon\lambda_1^2 + \frac{1}{\Delta}\left(\lambda_1\sqrt{2\epsilon_0}\left(2 + \frac{1}{\frac{5\lambda_1}{\Delta} - 1}\right)\prod_{i=1}^{\eta}\left(\frac{5\lambda_{i+1}}{\Delta}\right) - \frac{\lambda_1\sqrt{2\epsilon_0}}{\frac{5\lambda_1}{\Delta} - 1}\right).$$

And the rest of the result directly follows. $\qquad\square$

### B.3 PROOF OF LEMMA 3.1

*Proof.* To prove the above result, we consider two events $A_1 = \left\{\|\boldsymbol{y}\|_2 \geq \sqrt{\frac{1}{\epsilon}}\right\}$, $A_2 = \left\{|\boldsymbol{y}^\top \boldsymbol{v}| \leq \sqrt{\epsilon}\right\}$, then we can show that

$$\left\{|\boldsymbol{v}^\top \boldsymbol{x}| \leq \frac{1}{\sqrt{\epsilon}}\right\} \subset A_1 \cup A_2 \quad \Rightarrow \quad \mathbb{P}\left(|\boldsymbol{v}^\top \boldsymbol{x}| \leq \sqrt{\epsilon}\right) \leq \mathbb{P}(A_1) + \mathbb{P}(A_2).$$

And we use the tail bound for Chi-square given by (Laurent & Massart, 2000) to obtain that as $\epsilon < d^{-1}$,

$$\mathbb{P}(A_1) = \mathbb{P}\left(\|\boldsymbol{y}\|_2^2 \geq \epsilon^{-1}\right) \leq \exp\left(-C\epsilon^{-1}\right).$$

And similarly, consider the event $A_2$, note that $\boldsymbol{y}^\top \boldsymbol{v} \sim N(0,1)$, we use the cdf of the folded normal distribution to obtain that

$$\mathbb{P}(A_2) = \mathbb{P}\left(|\boldsymbol{v}^\top \boldsymbol{y}| \leq \sqrt{\epsilon}\right) = erf\left(\frac{\sqrt{\epsilon}}{\sqrt{2}}\right) = \frac{2}{\sqrt{\pi}}\left(\sqrt{\epsilon} - \frac{(\sqrt{\epsilon})^3}{3} + \frac{(\sqrt{\epsilon})^5}{10} - \frac{(\sqrt{\epsilon})^7}{42}\right) \leq \frac{\sqrt{\epsilon}}{\sqrt{\pi}}.$$

Then we obtain that

$$\mathbb{P}\Big(|\boldsymbol{v}^\top \boldsymbol{x}|\leq \sqrt{\epsilon}\Big) \leq \frac{\sqrt{\epsilon}}{\sqrt{\pi}} + \exp\left(-C\epsilon^{-1}\right).$$

Consider in total of $k$ independent random vectors $\boldsymbol{x}_1, \ldots, \boldsymbol{x}_k$, and arbitrary $k$ vectors $\boldsymbol{v}_1, \ldots, \boldsymbol{v}_k$, we can show that

$$\mathbb{P}\Big(\exists i \text{ such that } \boldsymbol{x}_i^\top \boldsymbol{v}_i \leq \epsilon\Big) \leq k\mathbb{P}\Big(\boldsymbol{x}_1^\top \boldsymbol{v}_1 \leq \epsilon\Big) \leq \frac{k\sqrt{\epsilon}}{\sqrt{\pi}} + k\exp(-C\epsilon^{-1}).$$

$\square$

### B.4 PROOF OF LEMMA B.1

**Lemma B.1.** *Assume that the correlation matrix $\boldsymbol{X}\boldsymbol{X}^\top$ has eigenvalues $\lambda_1 > \lambda_2 > \ldots > \lambda_k$. Assume that the eigenvectors are given by $\boldsymbol{v}_1, \boldsymbol{v}_2, \ldots, \boldsymbol{v}_n$ and the eigenvalues satisfy $\inf_{i \neq j}|\lambda_i - \lambda_j|= \Delta$. Then, given that the estimate for the first $\tau$ eigenvectors satisfy $\boldsymbol{v}_i^\top \widehat{\boldsymbol{v}}_i \geq 1 - \epsilon_i$ and the eigenvalues satisfy $|\lambda_i - \widehat{\lambda}_i|\leq \delta_i$, the principle eigenvector of $\boldsymbol{X}\boldsymbol{X}^\top - \sum_{i=1}^\tau \widehat{\lambda}_i \widehat{\boldsymbol{v}}_i \widehat{\boldsymbol{v}}_i^\top$ denoted by $\tilde{\boldsymbol{v}}_{\tau+1}$ satisfies*

$$\|\tilde{\boldsymbol{v}}_{\tau+1} - \boldsymbol{v}_{\tau+1}\|_2\leq \frac{\max_{i\in[\tau]} \delta_i + \sum_{i=1}^\tau \sqrt{8}\lambda_i\sqrt{\epsilon_i}}{\Delta}.$$

*Alternatively, we can also show that the eigenvector $\widehat{\boldsymbol{v}}_{\tau+1}$ returned by power method with $k = \frac{\log(1/\epsilon_0\delta)}{2\epsilon_0}$ that is initialized by satisfies*

$$\widehat{\boldsymbol{v}}_{\tau+1}^\top \boldsymbol{v}_{\tau+1} \geq 1 - \epsilon_{\tau+1} := 1 - \frac{1}{2}\Big(\frac{\max_{i\in[\tau]} \delta_i + \sum_{i=1}^\tau \sqrt{8}\lambda_i\sqrt{\epsilon_i}}{\Delta} + \sqrt{2\epsilon_0}\Big)^2,$$

*Proof.* Our proof is given by inductive arguments. Consider our obtained estimates $\{\widehat{\boldsymbol{v}}_i\}_{i\in[k]}$ for the eigenvectors $\{\boldsymbol{v}_i\}_{i\in[k]}$ satisfy

$$\boldsymbol{v}_i^\top \widehat{\boldsymbol{v}}_i \geq 1 - \epsilon_i \qquad \forall i \in [\tau], \qquad |\lambda_i - \widehat{\lambda}_i|\leq \delta_i.$$

We note that for the eigenvectors, we have for a vector $\boldsymbol{v}_0$,

$$\|\boldsymbol{v}_i\boldsymbol{v}_i^\top - \widehat{\boldsymbol{v}}_i\widehat{\boldsymbol{v}}_i^\top\|_2 = \sup_{\boldsymbol{v}_0\in\mathbb{S}^{d-1}} \boldsymbol{v}_0^\top(\boldsymbol{v}_i\boldsymbol{v}_i^\top - \widehat{\boldsymbol{v}}_i\widehat{\boldsymbol{v}}_i^\top)\boldsymbol{v}_0 = \sup_{\boldsymbol{v}_0\in\mathbb{S}^{d-1}} (\boldsymbol{v}_0^\top \boldsymbol{v}_i)^2 - (\boldsymbol{v}_0^\top \widehat{\boldsymbol{v}}_i)^2$$

$$= \sup_{\boldsymbol{v}_0\in\mathbb{S}^{d-1}} (\boldsymbol{v}_0^\top(\boldsymbol{v}_i - \widehat{\boldsymbol{v}}_i))(\boldsymbol{v}_0^\top(\boldsymbol{v}_i + \widehat{\boldsymbol{v}}_i^\top))$$

$$\leq 2\|\boldsymbol{v}_i - \widehat{\boldsymbol{v}}_i\|_2= 2\sqrt{\|\boldsymbol{v}_i - \widehat{\boldsymbol{v}}_i\|_2^2} = 2\sqrt{\|\boldsymbol{v}_i\|_2^2+\|\widehat{\boldsymbol{v}}_i\|_2^2-2\boldsymbol{v}_i^\top\widehat{\boldsymbol{v}}_i} = 2\sqrt{2\epsilon_i}.$$

Then, we can show by the subadditivity of the spectral norm,

$$\Big\|\boldsymbol{X}\boldsymbol{X}^\top - \sum_{i=1}^\tau \widehat{\lambda}_i\widehat{\boldsymbol{v}}_i\widehat{\boldsymbol{v}}_i^\top\Big\|_2 = \Big\|\sum_{i=1}^k \lambda_i\boldsymbol{v}_i\boldsymbol{v}_i^\top - \sum_{i=1}^\tau \widehat{\lambda}_i\widehat{\boldsymbol{v}}_i\widehat{\boldsymbol{v}}_i^\top\Big\|_2$$

$$\leq \Big\|\sum_{i=1}^k \lambda_i\boldsymbol{v}_i\boldsymbol{v}_i^\top - \sum_{i=1}^\tau \lambda_i\widehat{\boldsymbol{v}}_i\widehat{\boldsymbol{v}}_i^\top\Big\|_2 + \Big\|\sum_{i=1}^\tau \delta_i\widehat{\boldsymbol{v}}_i\widehat{\boldsymbol{v}}_i^\top\Big\|_2$$

$$\leq \Big\|\sum_{i=\tau+1}^k \lambda_i\boldsymbol{v}_i\boldsymbol{v}_i^\top\Big\|_2 + \Big\|\sum_{i=1}^\tau \delta_i\widehat{\boldsymbol{v}}_i\widehat{\boldsymbol{v}}_i^\top\Big\|_2 + \Big\|\sum_{i=1}^\tau \lambda_i\boldsymbol{v}_i\boldsymbol{v}_i^\top - \sum_{i=1}^\tau \lambda_i\widehat{\boldsymbol{v}}_i\widehat{\boldsymbol{v}}_i^\top\Big\|_2$$

$$\leq \lambda_{\tau+1} + \max_{i\in[\tau]}\delta_i + \Big\|\sum_{i=1}^\tau \lambda_i(\boldsymbol{v}_i\boldsymbol{v}_i^\top - \widehat{\boldsymbol{v}}_i\widehat{\boldsymbol{v}}_i^\top)\Big\|_2$$

$$\leq \lambda_{\tau+1} + \max_{i\in[\tau]}\delta_i + \sum_{i=1}^\tau \lambda_i\Big\|\boldsymbol{v}\boldsymbol{v}_i^\top - \widehat{\boldsymbol{v}}_i\widehat{\boldsymbol{v}}_i^\top\Big\|_2$$

$$\leq \lambda_{\tau+1} + \max_{i\in[\tau]}\delta_i + \sum_{i=1}^\tau \sqrt{8}\lambda_i\sqrt{\epsilon_i}.$$

By similar argument, we can also show that

$$\left\|\boldsymbol{X}\boldsymbol{X}^\top - \sum_{i=1}^\tau \widehat{\lambda}_i \widehat{\boldsymbol{v}}_i \widehat{\boldsymbol{v}}_i^\top\right\|_2 \geq \lambda_{\tau+1} - \max_{i\in[\tau]}\delta_i - \sum_{i=1}^\tau \sqrt{8}\lambda_i\sqrt{\epsilon_i}.$$

To study the convergence of the eigenvectors, we notice that by Davis-Kahan Theorem by (Yu et al., 2015) we can show that the principle eigenvector $\tilde{\boldsymbol{v}}_{\tau+1} = \arg\max_{\boldsymbol{v}\in\mathbb{S}^{d-1}}$ satisfies

$$\|\tilde{\boldsymbol{v}}_{\tau+1} - \boldsymbol{v}_{\tau+1}\|_2 \leq \frac{\left|\left\|\boldsymbol{X}\boldsymbol{X}^\top - \sum_i^\tau \widehat{\lambda}_i \widehat{\boldsymbol{v}}_i \widehat{\boldsymbol{v}}_i^\top\right\| - \lambda_{\tau+1}\right|}{\max\{|\lambda_{\tau+1}-\lambda_\tau|, |\lambda_{\tau-1}-\lambda_\tau|\}} \leq \frac{\max_{i\in[\tau]}\delta_i + \sum_{i=1}^\tau \sqrt{8}\lambda_i\sqrt{\epsilon_i}}{\Delta}.$$

Consider the eigenvector returned by the power method, we can show by the subadditivity of $L_2$ norm, we obtain that $\|\widehat{\boldsymbol{v}}_{\tau+1} - \boldsymbol{v}_{\tau+1}\|_2 \leq \|\tilde{\boldsymbol{v}}_{\tau+1} - \widehat{\boldsymbol{v}}_{\tau+1}\|_2 + \|\tilde{\boldsymbol{v}}_{\tau+1} - \boldsymbol{v}_{\tau+1}\|_2 \leq \frac{\max_{i\in[\tau]}\delta_i + \sum_{i=1}^\tau \sqrt{8}\lambda_i\sqrt{\epsilon_i}}{\Delta} + \sqrt{2\epsilon_0}$

$$\widehat{\boldsymbol{v}}_{\tau+1}^\top \boldsymbol{v}_{\tau+1} = \frac{1}{2}\left(2 - \|\boldsymbol{v}_{\tau+1} - \widehat{\boldsymbol{v}}_{\tau+1}\|_2^2\right) \geq \frac{1}{2}\left(2 - (\|\tilde{\boldsymbol{v}}_{\tau+1} - \widehat{\boldsymbol{v}}_{\tau+1}\|_2 + \|\boldsymbol{v}_{\tau+1} - \tilde{\boldsymbol{v}}_{\tau+1}\|_2)^2\right)$$

$$= 1 - \frac{1}{2}\left(\frac{\max_{i\in[\tau]}\delta_i + \sum_{i=1}^\tau \sqrt{8}\lambda_i\sqrt{\epsilon_i}}{\Delta} + \sqrt{2\epsilon_0}\right)^2.$$

Moreover, consider the estimate of the eigenvalue, we have

$$\left\|\left(\boldsymbol{X}\boldsymbol{X}^\top - \sum_{i=1}^\tau \widehat{\lambda}_i \widehat{\boldsymbol{v}}_i \widehat{\boldsymbol{v}}_i^\top\right)\widehat{\boldsymbol{v}}_{\tau+1}\right\|_2$$

$$\leq \left\|\left(\boldsymbol{X}\boldsymbol{X}^\top - \sum_{i=1}^\tau \lambda_i \boldsymbol{v}_i \boldsymbol{v}_i^\top\right)\widehat{\boldsymbol{v}}_{\tau+1}\right\|_2 + \left\|\sum_{i=1}^\tau \lambda_i \boldsymbol{v}_i \boldsymbol{v}_i^\top - \sum_{i=1}^\tau \widehat{\lambda}_i \widehat{\boldsymbol{v}}_i \widehat{\boldsymbol{v}}_i^\top\right\|_2$$

$$\leq \left\|\left(\boldsymbol{X}\boldsymbol{X}^\top - \sum_{i=1}^\tau \lambda_i \boldsymbol{v}_i \boldsymbol{v}_i^\top\right)\widehat{\boldsymbol{v}}_{\tau+1}\right\|_2 + \left\|\sum_{i=1}^\tau \lambda_i \boldsymbol{v}_i \boldsymbol{v}_i^\top - \sum_{i=1}^\tau \lambda_i \widehat{\boldsymbol{v}}_i \widehat{\boldsymbol{v}}_i^\top\right\|_2 + \left\|\sum_{i=1}^\tau \left(\lambda_i - \widehat{\lambda}_i\right)\widehat{\boldsymbol{v}}_i \widehat{\boldsymbol{v}}_i^\top\right\|_2$$

$$\leq \left\|\left(\boldsymbol{X}\boldsymbol{X}^\top - \sum_{i=1}^\tau \lambda_i \boldsymbol{v}_i \boldsymbol{v}_i^\top\right)\boldsymbol{v}_{\tau+1}\right\|_2 + \left\|\left(\boldsymbol{X}\boldsymbol{X}^\top - \sum_{i=1}^\tau \lambda_i \boldsymbol{v}_i \boldsymbol{v}_i^\top\right)\right\|_2 \|\widehat{\boldsymbol{v}}_{\tau+1} - \boldsymbol{v}_{\tau+1}\|_2$$

$$+ \max_{i\in[\tau]}\delta_i + \sum_{i=1}^\tau \sqrt{8}\lambda_i\sqrt{\epsilon_i}$$

$$= \lambda_{\tau+1} + \lambda_{\tau+1}\left(\frac{\max_{i\in[\tau]}\delta_i + \sum_{i=1}^\tau \sqrt{8}\lambda_i\sqrt{\epsilon_i}}{\Delta} + \sqrt{2\epsilon_0}\right) + \max_{i\in[\tau]}\delta_i + \sum_{i=1}^\tau \sqrt{8}\lambda_i\sqrt{\epsilon_i}.$$

Therefore, by similar arguments, we can show that

$$\left|\left\|\left(\boldsymbol{X}\boldsymbol{X}^\top - \sum_{i=1}^\tau \widehat{\lambda}_i \widehat{\boldsymbol{v}}_i \widehat{\boldsymbol{v}}_i^\top\right)\widehat{\boldsymbol{v}}_{\tau+1}\right\|_2 - \lambda_{\tau+1}\right| \leq \frac{2\lambda_{\tau+1}}{\Delta}\left(\max_{i\in[\tau]}\delta_i + \sum_{i=1}^\tau \sqrt{8}\lambda_i\sqrt{\epsilon_i}\right) + \lambda_{\tau+1}\sqrt{2\epsilon_0}.$$

$\square$

### B.5   Proof of Lemma B.2

**Lemma B.2** (Approximation of norm by sum of Relu activations by Transformer networks)**.** *Assume that there exists a constant $C$ with $\|\boldsymbol{v}\|_2 \leq C$. There exists a multihead Relu attention layer with number of heads $M < \left(\frac{\overline{R}}{\underline{R}}\right)^d \frac{C(d)}{\epsilon^2}\log(1 + C/\epsilon)$ such that there exsits $\{\boldsymbol{a}_m\}_{m\in[M]} \subset \mathbb{S}^{N-1}$ and $\{c_m\}_{m\in[M]} \subset \mathbb{R}$ where for all $\boldsymbol{v}$ with $\overline{R} \geq \|\boldsymbol{v}\|_2 \geq \underline{R}$, we have*

$$\left|\sum_{m=1}^M c_m \sigma(\boldsymbol{a}_m^\top \boldsymbol{v}) - \frac{1}{\|\boldsymbol{v}\|_2} + 1\right| \leq \epsilon.$$

*Similarly, there exists a multihead Relu attention layer with number of heads $M \leq \overline{R}^{\frac{d}{2}} \frac{C(d)}{\epsilon^2} \log(1 + C/\epsilon)$, a set of vectors $\{\boldsymbol{b}_m\}_{m\in[M]} \subset \mathbb{S}^{N-1}$ and $\{d_m\}_{m\in[M]} \subset \mathbb{R}$ such that*

$$\left| \sum_{m=1}^{M} d_m \sigma(\boldsymbol{b}_m^\top \boldsymbol{v}) - \|\boldsymbol{v}\|_2^{1/2} + 1 \right| \leq \epsilon.$$

*Proof.* Consider a set $\mathsf{C}^d(\overline{R}) := \mathsf{B}_\infty^d(\overline{R}) \setminus \mathsf{B}_2^d(\underline{R})$, then it is not hard to check that given $\|\boldsymbol{v}\|_2 > C$ with some $C(d) > 0$ depending on $d$ such that we have

$$\sup_{\boldsymbol{v} \in \mathsf{C}^d(\overline{R})} \partial_{v_{j_1},\dots,v_{j_i} \in [d]} \left( \frac{1}{\|\boldsymbol{v}\|_2} \right) \leq \frac{C(d)}{\|\boldsymbol{v}\|_2^d} \leq \frac{C(d)}{\underline{R}^d}.$$

Therefore, consider the definition 5, we have $C_\ell = \left( \frac{\overline{R}}{\underline{R}} \right)^d C(d)$. Note that by proposition A.1 in (Bai et al., 2024) shows that for a function that is $(R, C_\ell)$ smooth with $R \geq 1$ is $(\epsilon_{approx}, R, M, C)$ approximable with $M \leq C(d)C_\ell \log(1 + C_\ell/\epsilon_{approx})/\epsilon_{approx}^2$, we complete the proof.

Then we consider the function $\|\boldsymbol{v}\|_2^{\frac{1}{2}}$, note that

$$\sup_{\boldsymbol{v} \in \mathsf{C}^d(\overline{R})} \partial_{v_{j_1},\dots,v_{j_i} \in [d]} \|\boldsymbol{v}\|_2^{\frac{1}{2}} \leq C \|\boldsymbol{v}\|_2^{-\frac{1}{2}} \leq C\overline{R}^{-\frac{1}{2}}.$$

And the rest of the proof follows similarly to the previous step. $\qquad\square$

## C  EXPERIMENTAL DETAILS

### C.1  SETUP.

We run all our experiments on RTX 2080 Ti GPUs. We use PyTorch to construct our models and training process. We use sklearn for data generation. A training process with 2k steps roughly takes 0.5 hours.

### C.2  DATA.

**Synthetic Dataset.**   For each $X_i \in \mathbb{R}^D$, we sample $Z_i \sim N(0, I) \in \mathbb{R}^D$. We then form $Z = [Z_1, \cdots, Z_N]$ and transform it using an invertible matrix $L \sim N(0, I) \in \mathbb{R}^{D \times D}$, yielding the desired training sample $X$. To speed up the training process, we set $N < D$ in all our experiment setting. With this design, the rank of the covariance matrix $X^T X$ is at most $N$, meaning there are at least $D - N$ zero eigenvalues. The eigenvectors corresponding to these zero eigenvalues are less meaningful. Thus, to ensure predictions focus on meaningful eigenvectors, we increase $N$ to 10 when predicting multiple eigenvectors. We also adjust the data generation process to ensure the magnititude of eigenvalue across different $D$ to be at a similar level.

**Real-world Dataset.**   For both the MNIST and FMNIST, we first normalize the images to zero mean. Next, we perform SVD to extract the top-$D$ principal components and project the data onto these components, reducing feature dimension to $D = 10, 20$, and use $N = 10, 50$ for eigenvalue and eigenvector prediction respectively. Last, we rescale the resulting matrix to ensure its magnititude is roughly the same level as training data (transformers are trained on synthetic data). The rescaling process is critical to transformers as some images after SVD contains entries large as $7e3$. This will largely degrade transformer's performance as it changes the input domain by a large margin.

### C.3  HYPERPARAMETERS.

We list the hyperparameters in our experiments as below (table 2). We separate the hyperparameters used in predicting (1) eigenvalues and single eigenvectors, and (2) multiple eigenvectors.

Table 2: Hyperparameters for Eigenvalue and Eigenvector Prediction.

| parameter | $N = 5$ | $N = 10$ | $N = 20$ |
|---|---|---|---|
| steps (eigenvalue) | 20k | 20k | 20k |
| steps (eigenvector) | 20k | 20k | 60k |
| learning rate | 1e-3 | 5e-3 | 5e-3 |
| Optimizer | Adam | Adam | Adam |
| batch size | 64 | 64 | 64 |
| number of layers | 3 | 3 | 3 |
| hidden dimension | 64 | 64 | 64 |
| number of heads | 2 | 2 | 2 |

### C.4  ADDITIONAL EXPERIMENTAL RESULTS

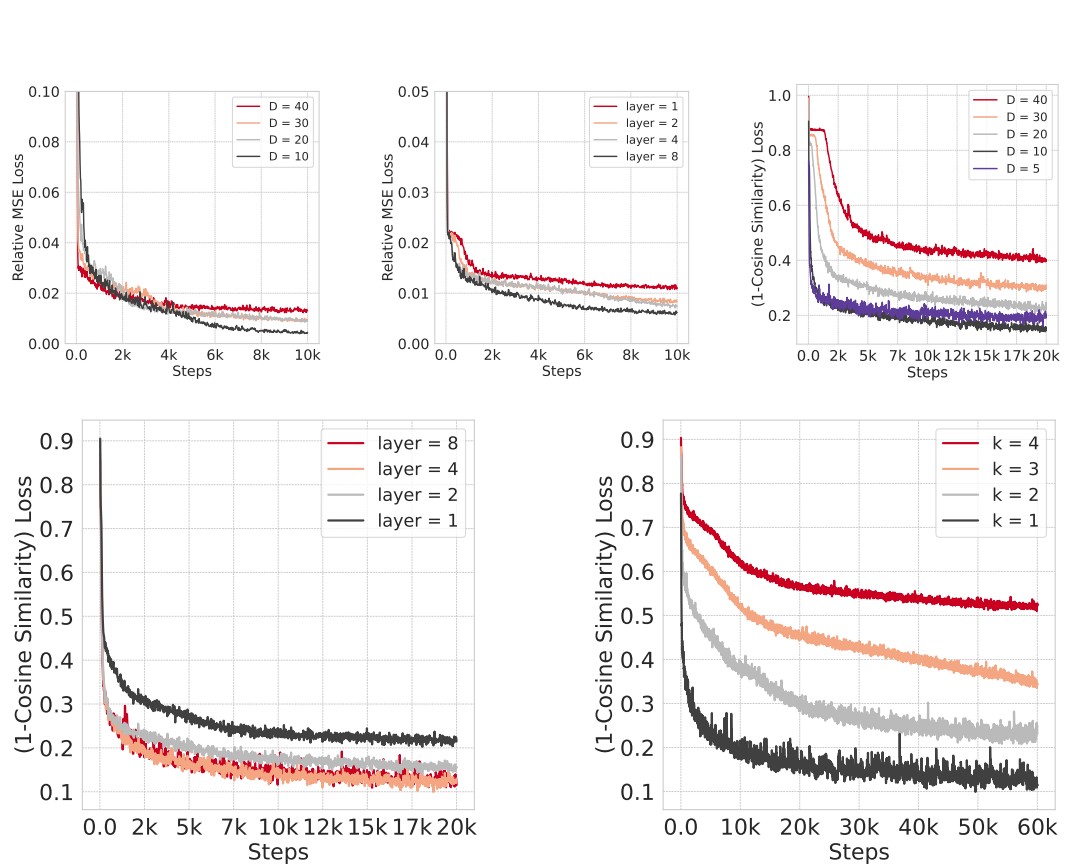

Figure 5: **Convergence Results on Eigenvalue, Eigenvector Prediction with Different Parameters.** *(1) Top left: Loss curve on eigenvalue prediction with different size of $D$ (2) Top middle: Loss curve on eigenvalue prediction with different number of layers (3) Top right: Loss curve on eigenvector prediction with different size of $D$ (4) Bottom left: Top right: Loss curve on eigenvector prediction with different number of layers (5) Bottom left: Loss curve on eigenvector prediction with different number of $k_{train}$* For (1), we observe that smaller $D$ is easiser for transformers as they present lower loss. For (2), we see that with more layers, transformers are also capable of predicting eigenvalues more accurately. For (3), transformers also predict eigenvectors better when $D$ is small. For (4), similar to (2), transformers with more layers shows improved performance. For (5), we want to highlight that the loss value is mainly affected by the fact that predicting 3rd or 4th eigenvectors are significantly harder, which contributes to higher loss value.

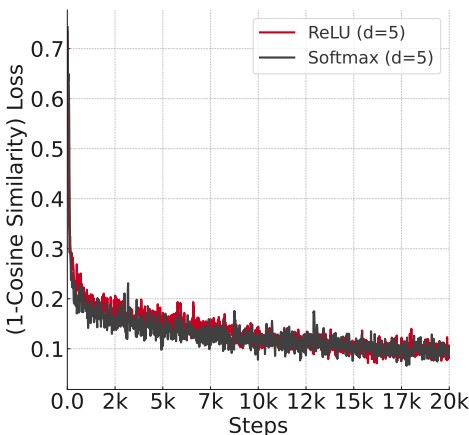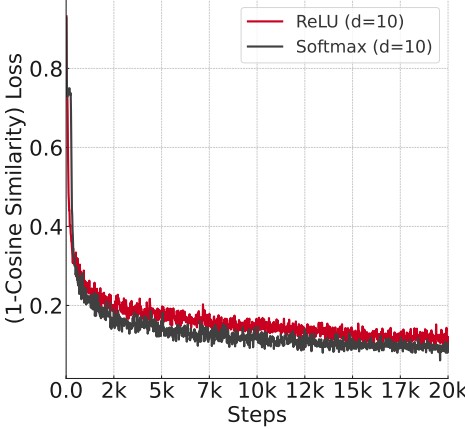

Figure 6: **Loss Curve Comparison between Softmax and ReLU Transformers (Top-1 Eigenvector Prediction).** *Left:* $D = 5$ *Right:* $D = 10$ We use a 3-layer, 2 head, 64 hidden dimension transformer to predict top-1 eigenvector across all experiments in this figure. An explanation for the superior performance of ReLU transformers is that the normalizing behavior of Softmax can potentially hinder the PCA process.

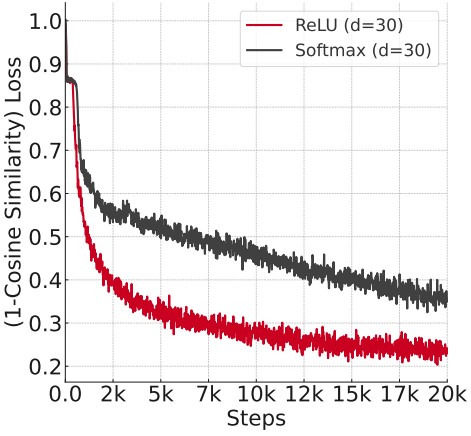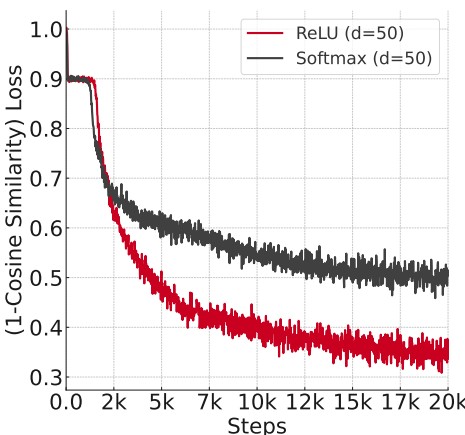

Figure 7: **Loss Curve Comparison between Softmax and ReLU Transformers (Top-1 Eigenvector Prediction).** *Left:* $D = 30$ *Right:* $D = 50$ We use a 3-layer, 2 head, 64 hidden dimension transformer to predict top-1 eigenvector across all experiments in this figure. We also observe that the performance gap enlarges as $D$ increases, likely because the difference between eigenvectors becomes larger with increasing $D$, making the normalizing nature of Softmax unsuitable for PCA.

