# OpenReview forum: "Can Transformers Perform PCA ?"
_ICLR.cc/2025/Conference — ICLR 2025 Conference Withdrawn Submission_

### Official Review · Reviewer_HAzx · 2024-10-28

**Soundness:** 2
**Presentation:** 1
**Contribution:** 2
**Rating:** 5
**Confidence:** 4

**Summary:**

The paper asks if a transformer can be used to calculate the top k eigenvectors of the covariance of an input data matrix. Towards this end, it uses the transformer to implement a power-method approach to this problem. The paper then analyzes the accuracy of this implementation (with respect to the true eigenvectors).

The question of what sort of data analysis transformers can do is interesting. While the paper makes some headway on this problem, it has some key limitations, discussed below:


1. For large enough transformers, why is it surprising that any analysis method, and in particular PCA can be implemented? Specifically, since the transformer described here can use self-attention (with relu) to calculate the covariance and then has enough layers to do the power iterations, why is it surprising that it can calculate PCA?


2. In Theorem 3.1., epsilon0 is used before being defined. You should write this more clearly/correctly by first quantifying over epsilon0 and then using it. But it still is strange that tau is upper bounded rather than lower bounded. Eg why not use tau=0. This theorem needs to be better written.


3. In the theory part ,d denotes the number of rows of X and D denotes the dimension after augmentation, but then in the experiments only D is mentioned as the dimension of the data. Is this a mistake? It seems like also in the experiments you would need both d and D.  Generally it is not clear what role P (augmentation matrix) plays in the experiments.


4. The standard method for calculating multiple eigenvectors is Lancoz, which I don’t think is what you are using. Have you considered using this instead?


5. There is a long line of work on implementing PCA with online rules such as Oja. It would be good to comment about this.


6. The paper is not very well written, with quite a few grammatical errors and typos (“One critical and most fundamental questions”, “Hence, practioners use various of methods”, “such that forward propagate along the it gives”).


7. What does “helps us screen out all the covariates” mean?


8. It seems like the use of ReLU is important for carrying out the covariance computation and that it would be hard to do with a softmax. Although there is some discussion of this, it seems like a significant restriction since softmax is much more broadly used.


9. I’m not sure how to understand the empirical results. They seem to mostly show that for smaller models it is harder to calculate PCA, which is perhaps not that surprising. Is

**Strengths:**

See above.

**Weaknesses:**

See above.

**Questions:**

See above.

---

> ### Author Response · Authors · 2024-12-04
> **Response to the Reviewer HAzx**
>
> We thank the reviewers for their constructive feedback. Here are our responses:
>
> Q1: *For large enough transformers, why is it surprising that any analysis method, and in particular PCA can be implemented? Specifically, since the transformer described here can use self-attention (with relu) to calculate the covariance and then has enough layers to do the power iterations, why is it surprising that it can calculate PCA?*
>
> A1: The important message is not that Transformers can perform PCA. Instead, we hope to use PCA as the simplest example to show that Transformers can learn to perform unsupervised algorithms from the pretraining phase. This is rather surprising since it might lead to our understanding of the significant inference capacities of LLMs on unsupervised learning tasks.
>
> Q2: *In Theorem 3.1., epsilon0 is used before being defined. You should write this more clearly/correctly by first quantifying over epsilon0 and then using it. But it still is strange that tau is upper bounded rather than lower bounded. Eg why not use tau=0. This theorem needs to be better written.*
>
> A2: We thank the reviewer for pointing out the typos in the theorem, which is revised in the full version of this paper.'
>
> Q3: *In the theory part ,d denotes the number of rows of X and D denotes the dimension after augmentation, but then in the experiments only D is mentioned as the dimension of the data. Is this a mistake? It seems like also in the experiments you would need both d and D. Generally it is not clear what role P (augmentation matrix) plays in the experiments.*
>
> A3: We thank the reviewer for pointing out this notation problem. In the experiments, we have $D=d$ since the auxiliary vectors in the theoretical part aren't used.
>
> Q4: *The standard method for calculating multiple eigenvectors is Lancoz, which I don’t think is what you are using. Have you considered using this instead?*
>
> A4: The reason for using the power method is to show the statistical guarantee of ERM. Other methods might lead to similar theoretical results since we just use this algorithm as a proof machine.
>
> Q5: *There is a long line of work on implementing PCA with online rules such as Oja. It would be good to comment on this.*
>
> A5: We thank the reviewer for pointing this out. However, the authors believe that this line of work is distantly related to ours due to the reason elaborated from A4. The algorithm in this work is just a proof machine and one can use whatever algorithm they like as long as they can construct the weight to approximate it. Finally they all comes to showing the properties of ERM.
>
> Q6: *The paper is not very well written, with quite a few grammatical errors and typos (“One critical and most fundamental questions”, “Hence, practioners use various of methods”, “such that forward propagate along the it gives”).*
>
> A6: We thank the reviewer for pointing this out. We corrected these grammatical errors in the full version of this paper.
>
> Q7: *What does “helps us screen out all the covariates” mean?*
>
> A7: By screening out all the covariates, we want to say that by constructing a proper left matrix $A$, one can recover identity (sub)matrix from $AH$ using this construction leaving no $X$ components in $AH$.
>
> Q8: *It seems like the use of ReLU is important for carrying out the covariance computation and that it would be hard to do with a softmax. Although there is some discussion of this, it seems like a significant restriction since softmax is much more broadly used.*
>
> A8: The biggest difficulty on the road is the lack of universal approximation results (from the approximation theory community) for Softmax functions. It is a $R^n\to R^m$ multivariate function and is significantly more difficult to analyze from the theoretical perspective. This is also why we use ReLU instead of Softmax.
>
> Q9: *I’m not sure how to understand the empirical results. They seem to mostly show that for smaller models it is harder to calculate PCA, which is perhaps not that surprising.*
>
> A9: We revised our simulation results extensively to show the logarithmic dependence between the loss and various parameters of the model in the full version of this work.

---

### Official Review · Reviewer_4cH8 · 2024-10-31

**Soundness:** 3
**Presentation:** 1
**Contribution:** 1
**Rating:** 3
**Confidence:** 2

**Summary:**

This paper explores the potential of transformer models to perform Principal Component Analysis (PCA) through a theoretical and empirical lens. Authors demonstrate that a pre-trained transformer can approximate the power method for PCA. The paper provides a rigorous proof of the transformer’s ability to estimate top eigenvectors and presents empirical evaluations on both synthetic and real-world datasets to validate these findings.

**Strengths:**

- The finding that transformers can effectively implement the power iteration method is intriguing and expands the known possibilities of transformers.
- The paper is theoretically rigorous, and the experiments on both synthetic and real-world datasets effectively support the proposed theoretical framework.

**Weaknesses:**

- The practical implications of the results are somewhat unclear. Does this result necessarily imply that transformers can perform PCA effectively on in-context examples?
- The novelty of this result is questionable; since even a linear autoencoder can approximate PCA, it is unclear why the fact that transformers with significantly more parameters can learn is surprising and valuable.
- The supervised pre-training phase seems unrealistic in practical applications, which makes the analysis and experimental results appear less impactful.
- The paper is challenging to follow due to unclear writing, which affects readability and the accessibility of its main ideas.

**Questions:**

- Lines 142-143: "Consider output dimension to be \tilde{D}, the…" – this sentence appears incomplete
- Several notations in the main body are not clearly defined. Implementing a more systematic notation or including a glossary would significantly improve readability.
- The notation L is used both for the number of layers and the loss function, which creates confusion.
- Could PCA not be formulated as an optimization problem and solved with gradient descent, using existing methods (e.g., [1])?

[1] Von Oswald, Johannes, et al. "Transformers learn in-context by gradient descent." *International Conference on Machine Learning*. PMLR, 2023.

---

> ### Author Response · Authors · 2024-12-04
> **Response to the reviewer 4cH8**
>
> We thank the reviewers for their constructive feedback. We give our response as the following:
>
> Q1: *The practical implications of the results are somewhat unclear. Does this result necessarily imply that transformers can perform PCA effectively on in-context examples?*
>
> A1: We hope to clarify that the problem setup in this work is **not** In-Context Learning. The major interest in the results provided by this work is to understand the strong capacities of Transformers in learning algorithms for unsupervised learning.
>
> Q2: *The novelty of this result is questionable; since even a linear autoencoder can approximate PCA, it is unclear why the fact that transformers with significantly more parameters can learn is surprising and valuable.*
>
> A2: As has been answered from the last question. The motivation of this work is to understand why pre-trained Transformers can perform unsupervised learning on unseen samples instead of trying to suggest that we should use Transformers to replace the existing PCA algorithms.
>
> Q3: *The supervised pre-training phase seems unrealistic in practical applications, which makes the analysis and experimental results appear less impactful.*
>
> A3: We hope to argue that existing LLMs are trained on enormous volumes of Internet data from various domains. This justifies that pretraining on large volumes of data is very crucial in improving the performance of the LLMs. Hence we believe that the pretraining phase is not unrealistic. Moreover, as this paper does not propose to use Transformers to do PCA, we believe our analysis and experimental results are contributing to understanding the Transformers. In addition, several works on in-context learning also include such a pre-training phase [1][2][3][4].
>
> [1] Li, Yingcong, et al. "Transformers as algorithms: Generalization and stability in in-context learning." International Conference on Machine Learning. PMLR, 2023.
>
> [2] Garg, Shivam, et al. "What can transformers learn in-context? a case study of simple function classes." Advances in Neural Information Processing Systems 35 (2022): 30583-30598.
>
> [3] Von Oswald, Johannes, et al. "Transformers learn in-context by gradient descent." International Conference on Machine Learning. PMLR, 2023.
>
> [4] Akyürek, Ekin, et al. "What learning algorithm is in-context learning? investigations with linear models." arXiv preprint arXiv:2211.15661 (2022).
>
> Q4: *The paper is challenging to follow due to unclear writing, which affects readability and the accessibility of its main ideas.*
>
> A4: We thank the reviewer for pointing this our and revised the writing in the full version of this paper.

---

### Official Review · Reviewer_mFmo · 2024-10-31

**Soundness:** 3
**Presentation:** 2
**Contribution:** 2
**Rating:** 5
**Confidence:** 4

**Summary:**

1. The paper shows how to do PCA using a forward pass through the transformers without going through power iteration
2. The error bounds are derived for Gaussian data, and they look sound
3. However, I believe the authors are trying a more difficult problem of estimating k eigenvectors simultaneously. For practical data, the method can solve the problem of finding the top eigenvector with sufficient accuracy. Instead, they can solve the easier problem of estimating the top one and then finding the next through successive elimination. It will still be fast enough for high dimension than power method

**Strengths:**

The maths in the paper is well-derived and sound. The method works well for the top eigenvector for datasets such as MNIST or F-MNIST. It has the potential to become useful. However, I believe the authors are trying a more difficult problem of estimating k eigenvectors simultaneously, and for practical data the method can solve find the top eigenvector with sufficient accurac. Instead they can solve the easier problem of estimating the top one, and then find the next ones through successive passes on X - \lambda ww'.

**Weaknesses:**

1. The error bound in Proposition 1 is proportional to d, the Gaussian dimension. I doubt the algorithm will work in high dimension

2. The experiments on synthetic data can include datasets for high dimensions (D). For D=50, people will simply use power iteration. The method's usefulness lies in whether it can predict the eigenvector/values for a high dimension, which the authors skipped for the synthetic data.

3. For the experiments on MNIST or F-MNIST, the cosine distance drops to 0.5 or below for k>1. This is concerning. If the transformer is trained to predict the top eigenvalue (say w) and eigenvector (say v) from X with sufficient accuracy, it can take another forward pass on X - w v v^T to predict the second pair. Why is the error for the eigenvector for k>1 so high? I believe the method needs some revision to be applicable

**Questions:**

Please see Weakness.

---

> ### Author Response · Authors · 2024-12-04
> **Response to the reviewer mFmo**
>
> We thank the reviewers for their constructive feedback. Here are our responses:
>
> Q1: *The error bound in Proposition 1 is proportional to d, the Gaussian dimension. I doubt the algorithm will work in high dimensions*
>
> A1: We agree with the reviewer that the dimension $d$ needs to stay below $\sqrt{n}$ to make the bound non-vacuous. This is because learning PCA is subject to the curse of dimensionality.
>
> Q2: *The experiments on synthetic data can include datasets for high dimensions (D). For D=50, people will simply use power iteration. The method's usefulness lies in whether it can predict the eigenvector/values for a high dimension, which the authors skipped for the synthetic data.*
>
> A2: In the full version of this work the authors provide a detailed comparison between the performance of using pre-trained Transformer and the power method in performing PCA. However, the authors would also like to clarify that the major goal of this work is not to suggest that one should use a pre-trained Transformer to perform PCA. Instead, this we aim to understand the strong learning abilities of Transformers through the lens of providing a theoretical analysis of the simplest unsupervised learning task of PCA.
>
> Q3: *For the experiments on MNIST or F-MNIST, the cosine distance drops to 0.5 or below for k>1. This is concerning. If the transformer is trained to predict the top eigenvalue (say w) and eigenvector (say v) from X with sufficient accuracy, it can take another forward pass on X - w v v^T to predict the second pair. Why is the error for the eigenvector for k>1 so high? I believe the method needs some revision to be applicable*
>
> A3: As the motivation of this work elaborated from the answer to the previous question, the authors simulate on the task of simultaneously recovering the top k eigenvectors rather than performing successive elimination. As has been suggested by the theoretical result in Theorem 1, the additional error imposed $k>1$ will signficantly affect the performance of Transformers.

---

### Official Review · Reviewer_b4mP · 2024-11-04

**Soundness:** 2
**Presentation:** 2
**Contribution:** 1
**Rating:** 3
**Confidence:** 3

**Summary:**

This work studies the problem of whether transformers can perform PCA. For this, they used a supervised setting where the outputs are principal components of the inputs. Inspired by the classical power iteration method, they construct the weights of a transformer model that approximately does PCA. For this, they assume that the eigenvalues are spaced out and bounded, and use classical results on random vectors to construct an auxiliary matrix they utilize to prove their bounds. Experiments on synthetic and real-life data weakly validate their studies. The target audience are people working on ML theory.

**Strengths:**

- The paper studies approximation capabilities of transformers from a theoretical perspective. This adds to a recent array of works which study the theoretical capabilities of transformers [1, 2, 3] and is potentially interesting.

- The experiments probe ablations of a few different parameters in both simulated and a couple of real-life datasets. While the observations are intuitive, it (weakly) validates some observations of the theory.

#### References:
- [1] Transformers Learn Shortcuts to Automata
- [2] Do LLMs dream of elephants (when told not to)? Latent concept association and associative memory in transformers.
- [3] (Un)interpretability of Transformers: a case study with bounded Dyck grammars

**Weaknesses:**

- While the question of transformers can perform PCA sounds interesting on the surface, I'm unable to gauge how interesting the bounds given here are. For one, we have universality results for neural networks that the authors cite, but do not seem to carefully compare against. Secondly, the bounds derived seem highly complicated and as the authors mention, it's not clear if they're tight. It's also not clear if they're useful or not, apart from being some generic generalization bounds. See also the question at the end.

- To continue the above point, this is more of a work on approximation of a transformer model to power iteration speficially, rather than to PCA. Lower bounds approximately validating their bounds would be useful here.

- The experiments seem a bit standalone and does not connect deeply to the theory, in particular the terms that arise in the loss. For example, the dimension D is hidden in the universal constant in remark 3, however it may be good to quantify the exact dependence and moreover, validate a version of it in the experiments.

- The paper seems hastily written, e.g. in L142, definition 3, The sentence defining \tilde{D} seems incomplete, L375 contains a missing citation, see also typos at the end.

**Questions:**

Some questions were raised above.

- I maybe misunderstanding something but aren't principal eigenvectors linearly related to the given vectors? If so, a linear matrix, instead of transformers, should suffice for this purpose?

#### Typos:

- L111: "along the it gives us"
- L132: "convinience"
- L233: "propogation"
- L308: "isotorpic"
- L721: "setps"
- ReLU, relu and Relu are used interchangeably.

---

> ### Author Response · Authors · 2024-12-04
> **Response to the reviewer b4mP**
>
> We thank the reviewers for their constructive feedback and advice. We summarize our response as follows:
>
> Q1: *For one, we have universality results for neural networks that the authors cite, but do not seem to carefully compare against.*
>
> A1: To the best of the authors' knowledge, no universality results for Transformers exist in the literature that covers the function mapping corresponding to PCA, which is a high dimensional vector-valued function. It is not hard to realize that universalities in approximating vector-valued functions are significantly more difficult and subtle due to the curse of dimensionality.
>
> Q2: *Secondly, the bounds derived seem highly complicated and as the authors mention, it's not clear if they're tight. It's also not clear if they're useful or not, apart from being some generic generalization bounds.*
>
> A2: We admit that the tightness might be difficult to achieve using Transformers since the approximation to the minimax optimal algorithm induces an extra error term. The authors believe that the bound yields insight into the strong learning capacities of Transformers in solving unsupervised learning problems by themselves.
>
> Q3: *To continue the above point, this is more of a work on the approximation of a transformer model to power iteration specifically, rather than to PCA. Lower bounds approximately validating their bounds would be useful here.*
>
> A3: We admit the lower bound is useful here. However, as mentioned in the answer to the previous question, the lower bound can be quite difficult to achieve since the minimax algorithm is further approximated by Transformers. The authors disagree that this work is on Transformers approximating power iteration. Power method approximation, on the other hand, is just a tool for the proof of the theoretical guarantees given by the ERM solution. One can replace the power method with other algorithms and achieve similar theoretical guarantees.
>
> Q4: *The experiments seem a bit standalone and do not connect deeply to the theory, in particular the terms that arise in the loss. For example, dimension D is hidden in the universal constant in remark 3, however, it may be good to quantify the exact dependence and validate a version of it in the experiments.*
>
> A4: We thank the reviewer for the suggestions on the connections between theory and simulations, which will be added to the full version of this work.
>
> Q4: *I maybe misunderstanding something but aren't principal eigenvectors linearly related to the given vectors? If so, a linear matrix, instead of transformers, should suffice for this purpose?*
>
> A4: In fact, the principle eigenvectors aren't linearly related to the given vectors and a linear matrix does not suffice for this work. As a reminder, we want to show that Transformers can learn the mapping from matrices to their eigenvectors through pretraining. Such mapping is highly nonlinear and cannot be learned by a linear NN.

---

### Official Review · Reviewer_NJ9K · 2024-11-09

**Soundness:** 3
**Presentation:** 1
**Contribution:** 2
**Rating:** 5
**Confidence:** 3

**Summary:**

The paper theoretically and empirically investigates whether one can use the Transformer architecture and supervised, in-context learning (ICL) to perform PCA, specifically the power method for computing top principle components.
The results build upon [Bai et al. (2024)](https://arxiv.org/abs/2306.04637)'s recent work on using Transformers and ICL to implement various ML algorithms (least-squares, ridge, lasso, and SGD).
The theoretical results provide an approximation bound on the PCs as well as a generalization error bound of $n^{-1/5}$.
Empirically, the authors show that the Transformer can approximate the first few principle components and the corresponding eigenvalues on Gaussian synthetic data and on real data (MNIST and Fashion-MNIST).

**Strengths:**

- The high-level problem statement is clear, timely, and potentially interesting. The theoretical and empirical results are novel.
- The theory portion of the paper appears to be rigorous (I did not check the proofs in detail).
- The experiments are generally reasonable and the results are consistent with the theory to some extent.

**Weaknesses:**

- The biggest issue for me is that the paper is not very clear about the significance of the results: _why_ is it important that Transformers can perform PCA? I don't imagine people using Transformers to compute PCA instead of the existing methods, so instead the results have to give meaningful insights on what it is about Transformers, or the ICL procedure, that allows them to perform PCA. I think there is a lot of missed potential in the discussion section to elaborate on this.
- A related question is: what is it about the task of PCA that sets it apart from other ML tasks in Bai et al. (2024), like least-squares, ridge, and lasso? Does the proof reveal any interesting insight about how either Transformers or ICL suit the specific task? Or is it just that any iterative algorithm can (in principle) be implemented in an ICL setting? How much would the performance degrade if I replace the Transformer model (partly or entirely) but keep the ICL framework? I think the paper has to be restructured in a way that some of the theoretical results are presented a bit more briefly and these questions are discussed (in words) in a lot more detail.
- The empirical results show that the principle components are not very accurate beyond the first few and/or in high dimensions. I think one thing that will make things clearer is if there were a baseline using just the power method on the same data, as this will clarify whether this is a limitation of the Transformer or the power method itself.
- Some of the conditions in the theorem are not well-explained/motivated in words, e.g., that the eigenvalues are distinct and that the L2 norm of the input is bounded (I don't see $B_X$ appearing in the resulting error bound). In particular, I think it's important to distinguish which conditions are necessary for the Power Method to work in the first place, and which are necessary for the Transformer to perform PCA.
- I'm not sure if I agree with the claim that "transformers are able to produce small error on predicting eigenvalues" on real data. The RMSE numbers are 10x larger than the synthetic case, and the eigenvectors are not similar for k >= 3. I think it would be more accurate to say that, on real data, the Transformer can approximate the first few eigenvectors and eigenvalues well, but not necessarily all of them. (This is not a weakness per se, but I think it's important to be clear about the limitations of the method.)

**Questions:**

- p. 4, text: I believe you means "symmetric" not "asymmetric" here? There are different terminologies being used between "principle eigenvectors", "left singular vectors", and of course "principle components" in PCA. My recommendation is to consistent terminology throughout the paper.
- p. 5, figure 1: what's the difference between blue and purple blocks?
- p. 6, remark 3: typo in "frist"
- p. 6, remark 5: what is the significance of the $n^{-1/5}$ rate specifically? How does one make sense of how good or bad this rate is?
- p. 7, line 375: typo "Figure ??", "differernt"
- p. 8, line 431: why exactly do these metrics (not loss functions, to be precise) match the intuition of eigenvalues/vectors?
- table 1: what is shown in parentheses?
- figures 2 & 4: why does the RMSE decrease as k increases for synthetic data, but increase for MNIST? Also, it may be a bit more intuitive to show the RMSE for individual components rather than the sum of the first k components.

---

> ### Author Response · Authors · 2024-12-03
> **Response to the reviewer NJ9K on the Theoretical Part 1**
>
> We thank the reviewers for their constructive feedback on this work. We also thank the reviewer for providing detailed advice on the notations and writings. We hope that the following response clarifies the misunderstandings and questions raised.
>
> Q1:  *Why is it important that Transformers can perform PCA?  I don't imagine people using Transformers to compute PCA instead of the existing methods, so instead the results have to give meaningful insights on what it is about Transformers, or the ICL procedure, that allows them to perform PCA. I think there is a lot of missed potential in the discussion section to elaborate on this.*
>
> A1: In principle, the meaning of the title Transformer being able to perform PCA suggests that Transformers are able to **learn** the procedure of PCA through retraining on a dataset where the matrices are mapped to their respective eigenvectors. Instead of showing that Transformers are better in performing PCA, we are more interested in understanding the capacity of learning unsupervised learning algorithms with PCA as the simplest example. In the literature of statistical unsupervised learning PCA is the foundational step for the spectral methods. Moreover, this work studies the unsupervised learning problem, which is significantly different from ICL where labels are presented in the input $H$.
>
> Q2: *What is it about the task of PCA that sets it apart from other ML tasks in Bai et al. (2024), like least-squares, ridge, and lasso?*
>
> A2: At a higher level of discussion, literature on Least-Squares, Ridge and Lasso focuses on the regression setup, which is under the supervised in context learning framework. PCA is instead an unsupervised learning problem, which has no label. At a lower level of discussion, PCA corresponds to a function that maps from matrices to vector spaces, which is significantly different from the setup considered in Bai et al 2024 where they consider functions that map from column vectors to labels (which is real valued).
>
> Q3: *Does the proof reveal any interesting insight about how either Transformers or ICL suit the specific task?*
>
> A3: The proof reveals that Transformers, being an auto-regressive model is able to leverage the correlation between the different columns in the input. As have mentioned from the previous question, ICL is under the supervised learning context whereas Transformers are under the unsupervised learning context.
>
> Q4: *Or is it just that any iterative algorithm can (in principle) be implemented in an ICL setting?*
>
> A4: This is a common misconception. Iterative algorithms can be implemented by Transformers ( not necessarily under ICL setup ) only if one can make a construction such that the iterations can be approximated through forward propagation along multi-layered Transformers. It is unknown if other iterative algorithms can be approximated without an explicit constructive proof.
>
> Q5: *How much would the performance degrade if I replace the Transformer model (partly or entirely) but keep the ICL framework?*
>
> A5: As has been mentioned in previous question the problem we focuses on is not under the ICL framework.
>
> Q6: *The empirical results show that the principle components are not very accurate beyond the first few and/or in high dimensions. I think one thing that will make things clearer is if there were a baseline using just the power method on the same data, as this will clarify whether this is a limitation of the Transformer or the power method itself.*
>
> A6: It is noted that our experiments are carried out under shallow Transformers. Intuitively our results should be compared with Power Method with linear number of iterations. It is also not hard to check that power method also requires significantly more steps to achieve convergence in high dimensional setup. Then the poor performance under the high dimensional setup is in fact not that suprising. Another problem is that we do not assume any sparsity assumptions here, which implies that the result will be restricted by the phenomenon of the curse of dimensionality. In the full version of this paper we present a comprehensive comparison between the two methods.

---

> > ### Author Response · Authors · 2024-12-03
> > **Response to the reviewer NJ9K on the Theoretical Part 2**
> >
> > Q7: *Some of the conditions in the theorem are not well-explained/motivated in words, e.g., that the eigenvalues are distinct and that the L2 norm of the input is bounded (I don't see  appearing in the resulting error bound).*
> >
> > A7: It is noted that in the theorem this bound is implicit in the $lambda_i$. And in the corollary the value $B_X$ appears in the upper bound.
> >
> > Q10: *What is the significance of the $N^{-1/5}$ rate specifically? How does one make sense of how good or bad this rate is?*
> >
> > A10: The significance of the $N^{-1/5}$ rate is that we can show polynomial decay of the $L_2$ norm. Since providing a lower bound for this problem requires us to provide a lower bound for the approximation error, which is very difficult to achieve, we are yet to know the tightness of this bound. However, a reference rate is $N^{1/2}$ for the generalization error bound. (which is typical in statistical learning theory, often referred to as the square-root $N$ rate)

---

> > > ### Author Response · Authors · 2024-12-04
> > > **Response to the reviewer NJ9K on the Simulation Part**
> > >
> > > Q1: *p. 5, figure 1: what's the difference between blue and purple blocks?*
> > >
> > > A1: The difference is the number of heads in the attention layer. There are only two heads in the blue blocks, and there are $M \gg 2$ heads in the purple blocks.
> > >
> > > Q2: *p. 8, line 431: why exactly do these metrics (not loss functions, to be precise) match the intuition of eigenvalues/vectors?*
> > >
> > > A2: The mean squared error metric evaluates how close the predicted eigenvalue is to the ground truth eigenvalue, providing a straightforward measure of accuracy. The relative version is preferred to ensure that the metric is independent of the scale of the eigenvalue. The scale can vary depending on the data dimension $d$, the number of sample $N$, and the $k$-th eigenvalue being predicted. Using the relative version prevents these variations from influencing the evaluation.
> > > For eigenvector prediction, we use cosine similarity as our metric. Cosine similarity only takes into account the alignment of two vectors while being independent of their magnitudes. This is ideal because eigenvectors can always be scaled by a scalar without affecting their direction.
> > >
> > > Q3: *table 1: what is shown in parentheses?*
> > >
> > > A3: Every evaluation point in our paper represents the average of three models trained with different random seeds. The values in parentheses in Table 1 indicate the standard deviation of the cosine similarity across these three runs.
> > >
> > > Q4: *figures 2 & 4: why does the RMSE decrease as k increases for synthetic data, but increase for MNIST? Also, it may be a bit more intuitive to show the RMSE for individual components rather than the sum of the first k components.*
> > >
> > > A4: For the first question, the RMSE increases for MNIST because the scale of the input differs significantly from the synthetic data used to train the Transformer. The parameter in the pre-trained transformer makes it hard to output the same trend on a real-world dataset. We verify this by dividing the input $X$ with various factors and observing that the trend of RMSE varies with the scaling factor. However, the RSME consistently remains within the range of $0.01\sim 0.15$.  For the second suggestion, we apologize for the confusion. The RMSE shown in Figures 2 and 4 is evaluated on individual components. The description in line 418 of the Metrics section is indeed unclear, and we have added a sentence to clarify this.

---

### Note · Authors · 2024-12-04

**Comment:**

Due to a conflict with the policy of the journal where the authors submit the full version of this work, the authors decide to withdraw. The authors thank the reviewers for their constructive feedback on the writing, presentations, and results of this work and will keep the audience updated with the full version of this work in the future.

**Withdrawal Confirmation:**

I have read and agree with the venue's withdrawal policy on behalf of myself and my co-authors.